# Rewind-to-Delete: Certified Machine Unlearning for Nonconvex Functions

**Siqiao Mu**
Department of Engineering Sciences and Applied Mathematics
Northwestern University
Evanston, IL 60208
siqiaomu2026@u.northwestern.edu

**Diego Klabjan**
Department of Industrial Engineering and Management Sciences
Northwestern University
Evanston, IL 60208
d-klabjan@northwestern.edu

## Abstract

Machine unlearning algorithms aim to efficiently remove data from a model without retraining it from scratch, in order to remove corrupted or outdated data or respect a user's "right to be forgotten." Certified machine unlearning is a strong theoretical guarantee based on differential privacy that quantifies the extent to which an algorithm erases data from the model weights. In contrast to existing works in certified unlearning for convex or strongly convex loss functions, or nonconvex objectives with limiting assumptions, we propose the first, first-order, black-box (i.e., can be applied to models pretrained with vanilla gradient descent) algorithm for unlearning on general nonconvex loss functions, which unlearns by "rewinding" to an earlier step during the learning process before performing gradient descent on the loss function of the retained data points. We prove $(\epsilon, \delta)$ certified unlearning and performance guarantees that establish the privacy-utility-complexity tradeoff of our algorithm, and we prove generalization guarantees for functions that satisfy the Polyak-Lojasiewicz inequality. Finally, we demonstrate the superior performance of our algorithm compared to existing methods, within a new experimental framework that more accurately reflects unlearning user data in practice.

## 1 Introduction

Machine unlearning, or data deletion from models, refers to the problem of removing the influence of some data from a trained model without the computational expenses of completely retraining it from scratch [6]. This research direction has become highly relevant in the last few years, due to increasing concern about user privacy and data security as well as the growing cost of retraining massive deep learning models on constantly updated datasets. For example, recent legislation that protects a user's "Right to be Forgotten," including the European Union's General Data Protection Regulation (GDPR), the California Consumer Privacy Act (CCPA), and the Canadian Consumer Privacy Protection Act (CPPA), mandate that users be allowed to request removal of their personal data, which may be stored in databases or memorized by models [38]. In addition, machine unlearning has practical implications for removing the influence of corrupted, outdated, or mislabeled data [28, 35].

The typical goal of machine unlearning algorithms is to yield a model that resembles the model obtained from a full retrain on the updated dataset after data is removed. This requirement is

39th Conference on Neural Information Processing Systems (NeurIPS 2025).

Table 1: Comparison of certified unlearning algorithms for convex and nonconvex functions, where $d$ is the dimension of the model parameters and $n$ is the size of the training dataset.

| Algorithm | Loss Function | Method | Storage | Black-box? |
|---|---|---|---|---|
| Newton Step[23] | Strongly convex | Second-order | $O(d^2)$ | $\times$ |
| Descent-to-Delete [34] | Strongly convex | First-order | $O(d)$ | $\checkmark$ |
| Langevin Unlearning [12] | Nonconvex | First-order | $O(d)$ | $\times$ |
| Constrained Newton Step [48] | Nonconvex | Second-order | $O(d^2)$ | $\checkmark$ |
| Hessian-Free Unlearning [37] | Nonconvex | Quasi-second-order | $O(nd)$ | $\times$ |
| **Our Work (R2D)** | **Nonconvex** | **First-order** | $O(d)$ | $\checkmark$ |

formalized in the concept of "certified unlearning," a strong theoretical guarantee motivated by differential privacy that probabilistically bounds the difference between the model weights returned by the unlearning and retraining algorithms [23]. However, algorithms satisfying certified unlearning need to also be practical. For example, retraining the model from scratch on the retained dataset is a trivial unlearning algorithm that provides no efficiency gain. On the other hand, unlearning is also provably satisfied if the learning and unlearning algorithm both output weights randomly sampled from the same Gaussian distribution, but this would yield a poorly performing model. Therefore, an ideal algorithm optimally balances data deletion, model accuracy, and computation, also known as the "privacy-utility-efficiency" tradeoff [32]. Moreover, because training from scratch is computationally expensive in the machine unlearning setting, we desire *black-box unlearning algorithms*, which can be applied to pretrained models and do not require training with the intention of unlearning later.

Most certified unlearning algorithms are designed for convex or strongly convex functions [23, 34, 38]. Relaxing the convexity requirement is challenging since nonconvex functions do not have unique global minima. Recently, there have been several works that provide certified unlearning guarantees for nonconvex functions. For example, [48] proposes a single-step Newton update algorithm with convex regularization followed by Gaussian perturbation, inspired by existing second-order unlearning methods such as [23, 38]. Similarly, [37] proposes a quasi-second-order method that exploits Hessian vector products to avoid directly computing the Hessian. Finally, [12] proposes a first-order method in the form of projected noisy gradient descent with Gaussian noise added at every step.

Prior work has focused on achieving theoretical unlearning guarantees in the nonconvex setting; however, practical unlearning algorithms should also be computationally efficient and convenient to use. First, [48] and [37] are both (quasi) second-order methods, but first-order methods that only require computing the gradient are more computationally efficient and require less storage. Second, [12, 37] are not black-box, since during the training process, [12] requires injecting Gaussian noise at every step and [37] requires storing a statistical vector for each data sample at each time step. These "white-box" algorithms require significant changes to standard learning algorithms, which hinders easy implementation. See Table 1 and Appendix E for an in-depth comparison with prior work.

In this work, we introduce "rewind-to-delete" (R2D), a first-order, black-box, certified unlearning algorithm for general nonconvex functions. Our learning algorithm consists of vanilla gradient descent steps on the loss function of the dataset followed by Gaussian perturbation to the model weights. To remove data from the model, our unlearning algorithm "rewinds" to a model checkpoint from earlier in the original training trajectory, followed by additional gradient descent steps on the *new* loss function and Gaussian perturbation. The checkpoint can be saved during the training period (which is standard practice), or it can be computed post hoc from a pretrained model via the proximal point method. We prove $(\epsilon, \delta)$ certified unlearning for our algorithm and provide theoretical guarantees that explicitly address the "privacy-utility-efficiency" tradeoff of our unlearning algorithm. Our algorithm is simple, easy to implement, and black-box, as it can be applied to a pretrained model as long as the model was trained with vanilla gradient descent.

We also analyze the case of nonconvex functions that satisfy the Polyak-Łojasiewicz (PL) inequality. The PL inequality is a weak condition that guarantees the existence of a connected basin of global minima, to which gradient descent converges at a linear rate [27]. This property allows us to derive empirical risk convergence and generalization guarantees, despite nonconvexity. PL functions are highly relevant to deep learning because overparameterized neural networks locally satisfy the PL condition in neighborhoods around initialization [30].

Finally, we empirically demonstrate the privacy-utility-efficiency tradeoff of our unlearning algorithm and its superior performance compared to other algorithms for nonconvex functions, within a chal-

lenging new framework in which the unlearned data is not independently and identically distributed (i.i.d.) to the training or test set. Most experiments select the unlearned samples either uniformly at random [48, 37] or all from one class [23, 20, 28], neither of which reflects unlearning in practice. For example, if we train a 10-class classifier and lose access to data from one class, we would not continue to use the same model on a 9-class dataset. In our approach, we instead train a model to learn some global characteristics about a real-world dataset derived from a collection of users. Upon unlearning, we remove data associated with a subset of the users, thereby simulating realistic unlearning requests and their impact on model utility. To assess the extent of unlearning, we adapt membership inference attack (MIA) methods [41, 9] to discriminate between the unlearned dataset and an *out-of-distribution* dataset, constructed separately from *users* not present in the training data. Our results demonstrate that R2D outperforms other certified and non-certified methods, showing a greater decrease in performance on the unlearned data samples and defending successfully against both types of membership inference attacks (Tables 2 and 3). In particular, compared to the white-box algorithm [37], R2D is up to *88* times faster during training, highlighting the advantage of our black-box approach (Tables 6 and 7). Ultimately, our first-order method enjoys the benefits of minimal computational requirements, easy off-the-shelf implementation, and competitive performance. While this work primarily addresses certified unlearning, our theoretical and empirical insights suggest that "rewinding" is a more appropriate baseline for comparison with new unlearning algorithms for deep neural networks, as opposed to to standard baselines like finetuning or gradient ascent.

Our contributions are as follows.

- We develop the first $(\epsilon, \delta)$ certified unlearning algorithm for nonconvex loss functions that is first-order and black-box.

- We prove theoretical unlearning guarantees that demonstrate the privacy-utility-efficiency tradeoff of our algorithm, allowing for controllable noise at the expense of privacy or computation. For the special case of PL loss functions, we obtain linear convergence and generalization guarantees.

- We empirically demonstrate the superior performance of our algorithm compared to both certified and non-certified unlearning algorithms, within a novel experimental framework that better reflects real-world scenarios for unlearning user data.

## 1.1 Related Work

**Differential privacy.** Differential privacy (DP) [15] is a well-established framework designed to protect individual privacy by ensuring that the inclusion or exclusion of any single data point in a dataset does not significantly affect the output of data analysis or modeling, limiting information leakage about any individual within the dataset. Specifically, a DP learning algorithm yields a model trained on some dataset that is probabilistically indistinguishable from a model trained on the same dataset with a data sample removed or replaced. The concept of $(\epsilon, \delta)$-privacy quantifies the strength of this privacy guarantee in terms of the privacy loss, $\epsilon$, and the probability of a privacy breach, $\delta$ [14]. This privacy can be applied during or after training by injecting controlled noise to the data, model weights [46, 49], gradients [1, 49], or objective function [8], in order to mask information about any one sample in the dataset. However, greater noise typically corresponds to worse model performance, leading to a trade-off between utility and privacy. The theory and techniques of differential privacy provide a natural starting point for the rigorous analysis of unlearning algorithms.

As observed in [46], "white-box" DP algorithms, which require code changes to inject noise at every training step, require additional development and runtime overhead and are challenging to deploy in the real world. Rather than adding noise at each iteration, [46] and [49] propose DP algorithms that only perturb the output after training, which is easier to integrate into standard development pipelines. In similar fashion, our proposed black-box unlearning algorithm can be implemented without any special steps during learning with gradient descent. We also do not inject noise at each iteration, and we only perturb at the end of training. The difference between our approach and [46, 49], however, is that our approach can accommodate the nonconvex case, leveraging model checkpointing to control the distance from the retraining trajectory.

**Certified unlearning.** The term "machine unlearning" was first coined by [6] to describe a deterministic data deletion algorithm, which has limited application to general optimization problems. In the following years, techniques have been developed for "exact unlearning," which exactly removes the

influence of data, and "approximate unlearning," which yields a model that is approximately close to the retrained model with some determined precision [47]. Our work focuses on the latter. Both [18] and [23] introduce a probabilistic notion of approximate unlearning, where the unlearning and retraining outputs must be close in distribution. Inspired by differential privacy, [23] introduces the definition of certified $(\epsilon, \delta)$ unlearning used in this work. Like DP algorithms, certified unlearning algorithms typically involve a combination of empirical risk minimization and noise injection to the weights or objective function, which can degrade model performance. Moreover, unlearning algorithms are also designed to reduce computation, leading to a three-way trade-off between privacy, utility, and complexity.

Certified unlearning has been studied for a variety of settings, including linear and logistic models [23, 25], graph neural networks [10], minimax models [31], and the federated learning setting [17], as well as convex models [38, 34, 42, 11] and nonconvex models [12, 37, 48]. These algorithms can be categorized as first-order methods that only require access to the function gradients [34, 11] or second-order methods that leverage information from the Hessian to approximate the model weights that would result from retraining [38, 42, 48, 37]. Our work is inspired by the "descent-to-delete" (D2D) algorithm [34], a first-order unlearning algorithm for strongly convex functions that unlearns by fine-tuning with gradient descent iterates on the loss function of the retained samples.

**Nonconvex unlearning.** There are also many approximate unlearning algorithms for nonconvex functions that rely on heuristics or weaker theoretical guarantees. For example, [4] proposes a "weak unlearning" algorithm, which considers indistinguishability with respect to model output space instead of model parameter space. Another popular algorithm for neural networks is SCRUB [28], a gradient-based algorithm that balances maximizing error on the unlearned data and maintaining performance on the retained data. An extension of SCRUB, SCRUB+Rewind, "rewinds" the algorithm to a point where the error on the unlearned data is "just high enough," so as to impede membership inference attacks. Furthermore, [20, 21] propose unlearning algorithms for deep neural networks, but they only provide a general upper bound on the amount of information retained in the weights rather than a strict certified unlearning guarantee. Additional approaches include subtracting out the impact of the unlearned data in each batch of gradient descent [22], gradient ascent on the loss function of the unlearned data [26], and retraining the last layers of the neural network on the retained data [19]. Ultimately, while the ideas of checkpointing, gradient ascent, and "rewinding" have been considered in other machine unlearning works, our algorithm combines these elements in a novel manner to obtain strong theoretical guarantees that prior algorithms lack.

Notably, virtually all unlearning papers implement a noiseless "finetuning" baseline method which is based on the D2D framework. This is usually because finetuning is first-order and easy to implement, even though D2D only has theoretical guarantees on strongly convex functions (which are impossible to extend to nonconvex settings) and finetuning does not perform well empirically on deep neural networks. In contrast, R2D is equally easy to implement, has theoretical guarantees on nonconvex functions, and empirically outperforms certified and non-certified methods. Therefore, our work provides strong support that rewinding instead of "descending" is a more appropriate baseline method for comparison.

## 2   Algorithm

Let $\mathcal{D} = \{z_1, ..., z_n\}$ be a training dataset of $n$ data points drawn independently and identically distributed from the sample space $\mathcal{Z}$, and let $\Theta$ be the (potentially infinite) model parameter space. Let $A : \mathcal{Z}^n \to \Theta$ be a (randomized) learning algorithm that trains on $\mathcal{D}$ and outputs a model with weight parameters $\theta \in \Theta$, where $\Theta$ is the (potentially infinite) space of model weights. Typically, the goal of a learning algorithm is to minimize $f_{\mathcal{D}}(\theta)$, the empirical loss on $\mathcal{D}$, defined as $f_{\mathcal{D}}(\theta) = \frac{1}{n} \sum_{i=1}^{n} f_{z_i}(\theta)$, where $f_{z_i}(\theta)$ represents the loss on the sample $z_i$.

Let us "unlearn" or remove the influence of a subset of data $Z \subset \mathcal{D}$ from the output of the learning algorithm $A(\mathcal{D})$. Let $\mathcal{D}' = \mathcal{D} \backslash Z$, and we denote by $U(A(\mathcal{D}), \mathcal{D}, Z)$ the output of an unlearning algorithm $U$. The goal of the unlearning algorithm is to output a model parameter that is probabilistically *indistinguishable* from the output of $A(\mathcal{D}')$. This is formalized in the concept of $(\epsilon, \delta)$-indistinguishability, which is used in the DP literature to characterize the influence of a data point on the model output [14].

**Definition 2.1.** [14, 34] Let $X$ and $Y$ be random variables over some domain $\Omega$. We say $X$ and $Y$ are $(\epsilon, \delta)$-indistinguishable if for all $S \subseteq \Omega$,

$$\mathbb{P}[X \in S] \leq e^\epsilon \mathbb{P}[Y \in S] + \delta,$$
$$\mathbb{P}[Y \in S] \leq e^\epsilon \mathbb{P}[X \in S] + \delta.$$

In the context of differential privacy, $X$ and $Y$ are the learning algorithm outputs on neighboring datasets that differ in a single sample. $\epsilon$ is the privacy loss or budget, which can be interpreted as a limit on the amount of information about an individual that can be extracted from the model, whereas $\delta$ accounts for the probability that these privacy guarantees might be violated. Definition 2.1 extends naturally to Definition 2.2, the definition of $(\epsilon, \delta)$ certified unlearning.

**Definition 2.2.** Let $Z \subset \mathcal{D}$ denote the data samples we would like to unlearn. Then $U$ is an $(\epsilon, \delta)$ certified unlearning algorithm for $A$ if for all such $Z$, $U(A(\mathcal{D}), \mathcal{D}, Z)$ and $A(\mathcal{D} \backslash Z)$ are $(\epsilon, \delta)$-indistinguishable.

Next, we describe our "rewind-to-delete" (R2D) algorithms for machine unlearning. The learning algorithm $A$ (Algorithm 1) performs gradient descent updates on $f_\mathcal{D}$ for $T$ iterations, and the iterate at the $T - K$ time step is saved as a checkpoint or computed post hoc via the proximal point algorithm (Algorithm 3). Then Gaussian noise is added to the final parameter $\theta_T$, and the perturbed parameter is used for model inference. When a request is received to remove the data subset $Z$, the checkpointed model parameter $\theta_{T-K}$ is loaded as the initial point of the unlearning algorithm $U$ (Algorithm 2). Then we perform $K$ gradient descent steps on the *new* loss function $f_{\mathcal{D}'}$ and add Gaussian noise again to the final parameter, using the perturbed weights for future model inference.

---

**Algorithm 1** $A$: R2D Learning Algorithm

**Require:** dataset $\mathcal{D}$, initial point $\theta_0 \in \Theta$
  **for** t = 1, 2, ..., T **do**
    $\theta_t = \theta_{t-1} - \eta \nabla f_\mathcal{D}(\theta_{t-1})$
  **end for**
  Save checkpoint $\theta_{T-K}$ **or** compute $\theta_{T-K}$ via Algorithm 3
  Sample $\xi \sim \mathcal{N}(0, \sigma^2 \mathbb{I}_d)$
  $\tilde{\theta} = \theta_T + \xi$
  Use $\tilde{\theta}$ for model inference
  Upon receiving an unlearning request, call Algorithm 2

---

**Algorithm 2** $U$: R2D Unlearning Algorithm

**Require:** dataset $\mathcal{D}'$, model checkpoint $\theta_{T-K}$
  $\theta_0'' = \theta_{T-K}$
  **for** t = 1, ..., K **do**
    $\theta_t'' = \theta_{t-1}'' - \eta \nabla f_{\mathcal{D}'}(\theta_{t-1}'')$
  **end for**
  Sample $\xi \sim \mathcal{N}(0, \sigma^2 \mathbb{I}_d)$
  $\tilde{\theta}'' = \theta_K'' + \xi$
  Use $\tilde{\theta}''$ for model inference

---

**Algorithm 3** Compute Checkpoint via Proximal Point Method

**Require:** datasets $\mathcal{D}$, model checkpoint $\theta_T$
  $w_0 = \theta_T$
  **for** t = 1, ..., K **do**
    $w_t = \arg \min_x \{-f_\mathcal{D}(x) + \frac{1}{2\eta} \|x - w_{t-1}\|^2\}$
  **end for**
  **return** $w_K$

---

When a model is trained without a checkpoint saved, we can still obtain a black-box unlearning algorithm by carefully undoing the gradient descent training steps via the proximal point method, outlined in Algorithm 3. We can solve for previous gradient descent iterates through the following implicit equation (2):

$$\theta_{t+1} = \theta_t - \eta \nabla f_\mathcal{D}(\theta_t) \tag{1}$$
$$\theta_t = \theta_{t+1} + \eta \nabla f_\mathcal{D}(\theta_t). \tag{2}$$

The backward Euler update in (2) is distinct from a standard gradient ascent step, which is a forward Euler method. Instead, we compute $\theta_t$ from $\theta_{t+1}$ by taking advantage of the connection between backward Euler for gradient flow and the proximal point method, an iterative algorithm for minimizing a convex function [33]. Let $g(\theta)$ denote a convex function. The proximal point method minimizes $g(\theta)$ by taking the proximal operator with parameter $\gamma$ of the previous iterate, defined as follows

$$\theta_{k+1} = prox_{g,\gamma}(\theta_k) = \arg \min_x \{g(x) + \frac{1}{2\gamma} \|x - \theta_k\|^2\}.$$

Although for our problem, $f$ is nonconvex, adding sufficient regularization produces a convex and globally tractable proximal point subproblem, stated in Lemma A.1. Therefore, by computing the proximal operator with respect to $-f(\theta)$, we can solve the implicit gradient ascent step.

**Lemma 2.1.** *Suppose $f(\theta)$ is continuously differentiable, $\theta_t$ is defined as in (2), and let $\eta < \frac{1}{L}$. Then $\theta_t = prox_{-f,\eta}(\theta_{t+1})$.*

If $\eta < \frac{1}{L}$, then due to strong convexity, we can solve the proximal point subproblem easily via gradient descent or Newton's method. Computationally, when $K \ll T$, the algorithm is comparable to other second-order unlearning methods that require a single Newton step. In addition, we only need to compute the model checkpoint once prior to unlearning requests, so this computation can be considered "offline" [25, 37]. In Appendix B.3, we test the ability of Algorithm 3 to reconstruct prior training steps in practice (Table 8), and we measure the additional computation time required (Tables 9 and 10). We measure the Euclidean distance between the original checkpoint and the reconstructed checkpoint, observing that we can quite faithfully reconstruct checkpoints for $K < T/2$, but beyond this point we encounter instability issues due to compounding approximation errors.

In general, our work shares similarities with Descent-to-Delete (D2D), which "descends" from the trained model $\theta_T$ instead of from an earlier checkpoint $\theta_{T-K}$. However, extending the D2D analysis beyond the strongly convex setting is impossible since it relies on the existence of a unique global minimum, which (i) attracts training trajectories and (ii) remains in a small neighborhood when the underlying loss function is changed. Neither (i) nor (ii) hold for the general nonconvex setting, where we may only converge to a local minima or saddle point. Our novel insight is we leverage rewinding instead of "descending" to bring the unlearned model closer to the retrained model by reversing the divergence in training trajectories caused by the unlearning bias.

## 3   Analyses

**The proofs of all theoretical results can be found in Appendix A.** In the following theorem, we establish the unlearning and performance guarantees for our algorithm on nonconvex functions. For nonconvex functions, gradient descent might converge to local minima or saddle points, so we measure the performance by the average of the gradient norm over iterates, a common DP performance metric for algorithms on nonconvex functions.

**Theorem 3.1.** *Let $\epsilon, \delta$ be fixed such that $0 < \epsilon \le 1$ and $\delta > 0$. Suppose for all $z \in \mathcal{Z}$, the loss function $f_z$ is L-Lipschitz smooth and the gradient is uniformly bounded by some constant $G$ so that $\|\nabla f_z(\theta)\| < G$ for all $\theta \in \Theta$. Let $\mathcal{D}$ denote the original dataset of size $n$, let $Z \subset \mathcal{D}$ denote the unlearned dataset of size $m$, and let $\mathcal{D}' = \mathcal{D} \backslash Z$ denote the retained dataset. Let the learning algorithm $A$ be initialized at $\theta_0$ and run for $T$ iterations with step size $\eta \le \min\{\frac{1}{L}, \frac{n}{2(n-m)L}\}$. Let the standard deviation $\sigma$ of the Gaussian noise be defined as*

$$\sigma = \frac{2mG \cdot h(K)\sqrt{2\log(1.25/\delta)}}{Ln\epsilon}, \tag{3}$$

*where $h(K)$ is a function that monotonically decreases to zero as $K$ increases from $0$ to $T$ defined by*

$$h(K) = ((1 + \frac{\eta Ln}{n-m})^{T-K} - 1)(1 + \eta L)^K.$$

*Then $U$ is an $(\epsilon, \delta)$-unlearning algorithm for $A$ with noise $\sigma$. In addition,*

$$\frac{1}{T}(\sum_{t=0}^{T-K-1} \|\nabla f_{\mathcal{D}'}(\theta_t)\|^2 + \sum_{t=0}^{K-1} \|\nabla f_{\mathcal{D}'}(\theta_t'')\|^2 + \mathbb{E}[\|\nabla f_{\mathcal{D}'}(\tilde{\theta}'')\|^2])$$

$$\le O(\sigma^2 + \frac{n}{T(n-m)} + \frac{(T-K-1)m}{T(n-m)}), \tag{4}$$

*where the expectation is taken with respect to the Gaussian noise added at the end of $U$, and where $O(\cdot)$ hides dependencies on $\eta$, $G$, and $L$.*

**Corollary 3.2.** *For fixed $\sigma$ and $\delta$, the dependence of $K$ on $\epsilon$ in (3), denoted as $K(\epsilon)$, is bounded as follows:*

$$K(\epsilon) \le (\log(1 + \frac{\eta Ln}{n-m}))^{-1} \log((1 + \frac{\eta Ln}{n-m})^T - \frac{\sigma Ln\epsilon}{2mG\sqrt{2\log(1.25/\delta)}}). \tag{5}$$

Equation (4) states that the average of the gradient norm squared of the initial $T - K$ learning iterates, the $K - 1$ unlearning iterates after, and the last perturbed iterate $\tilde{\theta}''$ decreases with increasing $T$ and $n$, indicating that the algorithm converges to a stationary point with small gradient norm. Corollary 3.2 provides an upper bound on $K(\epsilon)$, such that in practice we can choose $K$ equal to this bound to ensure the privacy guarantee is achieved.

The analysis relies on carefully tracking the distance between the unlearning iterates and gradient descent iterates on $f_{\mathcal{D}'}$. Like prior work in certified unlearning, our analysis relies on the Gaussian mechanism for differential privacy [14], which implies that as long as the distance between the trajectories is bounded, we can add a sufficient amount of Gaussian noise to make the algorithm outputs $(\epsilon, \delta)$-indistinguishable. We therefore can compute the noise level $\sigma$ required to achieve unlearning. For the utility guarantees, we leverage the fact that gradient descent steps on $f_{\mathcal{D}}$ also make progress on $f_{\mathcal{D}'}$ to obtain bounds that only depend on problem parameters and $\theta_0$.

Theorem 3.1 underscores the "privacy-utility-complexity" tradeoff between our measure of unlearning, $\epsilon$, noise, $\sigma$, and the number of unlearning iterations, $K$. By construction, $K < T$, so our algorithm is more efficient than retraining. We can pick larger $K$ such that the noise required is arbitrarily small at the expense of computation, and when $K = T$ our algorithm becomes a noiseless full retrain. Moreover, the standard deviation $\sigma$ inversely scales with the size of the dataset $n$, implying that unlearning on larger datasets require less noise. In contrast, [48] and [37] do not feature such data-dependent guarantees. In practice, following the guidelines in [48], we suggest first choosing $\sigma$ that preserves model utility (in our work, we choose $\sigma = 0.01$) and $K$ within the computational budget, such as $10\% \times T$. From there, one can compute the level of privacy achieved. Our experiments for various $\sigma$, including $\sigma = 0$, (Table 13 and 15) suggest that a small amount of rewinding or noise performs well empirically.

Theorem 3.1 applies to unlearning a batch of $m$ data samples, but our algorithm also accommodates sequential unlearning requests. If, after unlearning $m$ points, an additional $k$ unlearning requests arrive, we simply call the unlearning algorithm $M$ on the new retained dataset of size $n - m - k$. Notably, if the total number of unlearned data increases while $\sigma$ and $K$ stay constant, our unlearning guarantee worsens, which aligns with other results [23, 48, 12].

We can obtain faster convergence for Polyak-Lojasiewicz (PL) functions, which are functions that satisfy the PL inequality (6). Although PL functions can be nonconvex, they have a continuous basin of global minima, enabling both empirical and population risk bounds.

**Definition 3.3.** [27] For some function $f$, suppose it attains a global minimum value $f^*$. Then $f$ satisfies the PL inequality if for some $\mu > 0$ and all $x$,

$$\frac{1}{2}||\nabla f(x)||^2 \geq \mu(f(x) - f^*). \tag{6}$$

Although practical algorithms typically minimize the empirical risk, the ultimate goal of learning is to minimize the population risk $F$, in order to determine how well the model will generalize on unseen test data. We leverage results from [29] that relate the on-average stability bounds of algorithms on PL functions to their excess population risk.

**Corollary 3.3.** *Suppose the conditions of Theorem 3.1 hold and in addition, $f_{\mathcal{D}'}$ satisfies the PL condition (6) with parameter $\mu$. Let $F$ represent the population risk, defined as $F(\theta) = \mathbb{E}_{z \sim \mathcal{Z}}[f_z(\theta)]$, and let $F^*$ represent its global optimal value. Then we have*

$$\mathbb{E}[F(\tilde{\theta}'')] - F^* \leq L\sqrt{d}\sigma + \frac{2G^2}{(n-m)\mu}$$
$$+ \frac{L}{2\mu}(1-\eta\mu)^K[(1 - \frac{\eta\mu(n-m)}{n})^{T-K}(f_{\mathcal{D}'}(\theta_0) - f_{\mathcal{D}'}^*) + \frac{Gm(G+L\eta)}{\mu(n-m)}]$$

*where $\sigma$ is defined in (3) and the expectation is taken with respect to i.i.d. sampling of $\mathcal{D} \sim \mathcal{Z}$ and the Gaussian noise added at the end of $U$.*

Our performance guarantees demonstrate that more learning iterates $T$ correspond to better performance upon unlearning for fixed $\sigma$. For PL functions, the utility converges faster with increasing $T$ than for the general nonconvex case.

# 4 Experiments

See Appendix B for an in-depth review of experimental details, including hyperparameters and hardware (B.1), implementation details of MIAs (B.1.1) and baseline methods (B.1.2), computation time experiments (B.2), proximal point method (Algorithm 3) experiments (B.3), additional numerical results (B.5), and the GitHub repository.

## 4.1 Setup

**Experimental Framework.** We test Algorithms 1 and 2 in a novel setting where the unlearned dataset is not i.i.d. to the training or test dataset. For all experiments, we train a binary classifier with the cross-entropy loss function over a dataset that is naturally split among many different users. To test unlearning, we remove the data associated with a subset of the users (1%-2% of the data) and observe the impact on the original classification task. Our framework better reflects unlearning in practice, where users may request the removal of their data but they do not each represent a class in the model. This stands in contrast to current experimental approaches that unlearn data from a selected class [23, 20, 28], or randomly select samples uniformly from the dataset to unlearn [48, 37].

**Datasets and Models.** We consider two real-world datasets and neural network models with highly *nonconvex* loss functions. For small-scale experiments, we train a multilayer perceptron (MLP) with 3 hidden layers to perform classification on the eICU dataset, a large multi-center intensive care unit (ICU) database consisting of tabular data on ICU admissions [36]. Each patient is linked with 1-24 hospital stays. We predict if the length of a hospital stay of a patient is longer or shorter than a week using the intake variables of the Acute Physiology Age Chronic Health Evaluation (APACHE) predictive framework, including blood pressure, body temperature, and age. For unlearning, we remove a random subset of patients and their corresponding data. For large-scale experiments, we consider a subset of the VGGFace2 dataset, which is composed of approximately $9,000$ celebrities and their face images from the internet [5]. We apply the MAAD-Face annotations from [44] to label each celebrity as male or female, and sample a class balanced dataset of 100 celebrities to form the Lacuna-100 dataset as described in [20]. We train a ResNet-18 model to perform binary gender classification. For unlearning, we remove a random subset of the celebrities and their face images.

**Implementation.** During the learning process, we train the model and save checkpoints every 10 epochs. Upon unlearning, we revert to earlier checkpoints and train on the new loss function. For each version of R2D, we compute the "rewind percent" as $\frac{K}{T} \times 100\%$, or the number of unlearning steps $K$ as a fraction of the number of original training steps $T$. Due to the computational demands of full-batch gradient descent, we implement our algorithms using mini-batch gradient descent with a very large batch size (2048 for eICU and 512 for Lacuna-100). For $(\epsilon, \delta)$-unlearning, we utilize the bound derived in [3] to calibrate the noise for $\epsilon > 1$, and we estimate the values $L$ and $G$ using sampling approaches outlined in Appendix B.1.

**Unlearning Metrics.** To empirically evaluate the unlearned model, we apply membership inference attacks (MIA) to attempt to distinguish between unlearned data and data that has never been in the training dataset. A lower Area Under the Receiver Operating Characteristic Curve (AUC) of the membership attack model corresponds to less information retained in the model and more successful unlearning. We employ both the classic MIA [41], that only considers the output of the model after unlearning, and an advanced attack tailored to the unlearning setting (MIA-U) [9], that compares the output of the original model and the unlearned model. These attacks are typically performed on i.i.d. datasets, but we adapt them to our setting by constructing an *out-of-distribution* (OOD) dataset representing data from *users* absent from the training data. For Lacuna-100, we construct an OOD dataset using an additional 100 users from the VGGFace2 database. For eICU, we construct the OOD dataset from data samples in the test set belonging to users not present in the training set.

We also consider the performance (AUC) on the retain set, unlearned set, and test set (denoted as $\mathcal{D}_{retain}, \mathcal{D}_{unlearn}, \mathcal{D}_{test}$) of both the original trained model and the model after unlearning (Table 2). A decrease in performance on $\mathcal{D}_{unlearn}$ suggests the model is losing information about the data.

**Baseline Methods.** We compare against two other certified unlearning algorithms for nonconvex functions: **Constrained Newton Step (CNS)** [48], a black-box algorithm which involves a single Newton step within a constrained parameter set, and **Hessian-Free Unlearning (HF)** [37], a white-box algorithm which involves demanding pre-computation and storage of data influence vectors

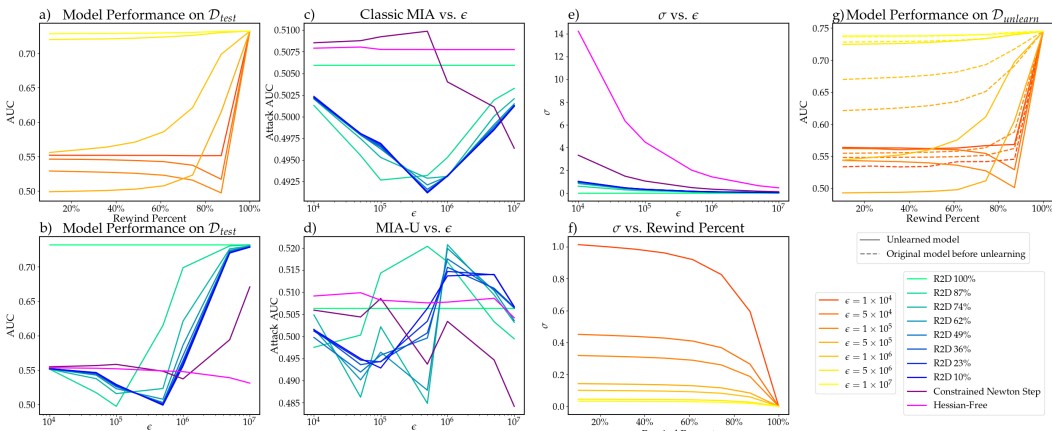

Figure 1: Privacy-utility-complexity tradeoff of R2D compared to other certified unlearning methods (Constrained Newton Step and Hessian-Free method) on the eICU dataset. In Figures 1a, 1f, and 1g, we plot against the rewind percent, computed as $\frac{K}{T} \times 100\%$.

during training, allowing unlearning via simple vector addition later. Both methods also involve a single Gaussian perturbation at the end to achieve $(\epsilon, \delta)$-unlearning. We do not implement the white-box algorithm from [12] because, as stated in their work, "the non-convex unlearning bound... is not tight enough to be applied in practice due to its exponential dependence on various hyperparameters." For all baselines methods, we use the hyperparameters reported in the original papers that align with our settings. However, we note that certified unlearning algorithms can be highly sensitive to hyperparameter choices, so for a fair comparison we also consider a flat noise amount of $\sigma = 0.01$, following the precedent established in [37, 48, 23]. Moreover, this allows us to compare against several state-of-the-art baseline methods which do not have theoretical unlearning guarantees but can be evaluated empirically: **Finetune**, where the model is fine-tuned on the retained dataset, **Fisher Forgetting** [20], which selectively perturbs weights via the Fisher information matrix, and **SCRUB** [28]. Unlike the certified unlearning methods, these algorithms do not require any specific training procedure, so we apply them to the same original trained model used for R2D.

### 4.2 Results

In Figure 1, we compare R2D with varying $\epsilon$ and $K$ to other certified unlearning algorithms on the eICU dataset. Figures 1a and 1b demonstrate that model performance (AUC) on $\mathcal{D}_{test}$ improves with more rewinding and larger $\epsilon$, which correspond to smaller $\sigma$ as shown from the result derived in (3). Figure 1b shows that R2D has a superior privacy-utility tradeoff, with better performance on $\mathcal{D}_{test}$ due to a smaller noise requirement $\sigma$ that scales more advantageously with $\epsilon$, as shown in Figure 1e. In addition, for most values of $\epsilon$, R2D is able to defend more successfully against the membership attacks, demonstrated by lower MIA scores in Figure 1c and 1d. Finally, Figure 1g displays the performance of the model before and after unlearning on $\mathcal{D}_{unlearn}$, with the decrease in performance after unlearning indicating that the model is losing information about the samples.

We further compare R2D to certified and non-certified algorithms in Tables 2 and 3, with the full numerical results available in Tables 12 and 14 in Appendix B.5. Table 2 displays the model AUC on $\mathcal{D}_{retain}$, $\mathcal{D}_{unlearn}$, and $\mathcal{D}_{test}$ before and after unlearning, and Figure 2 plots these values over rewind percent. We note that there are different "original models" because the certified baseline methods (HF, CNS) require different training procedures. On both datasets, R2D displays a significantly greater performance drop on $\mathcal{D}_{unlearn}$ compared to non-certified methods, while outperforming CNS on eICU and both HF and CNS on Lacuna-100. We observe that as expected, more rewinding decreases the utility on $\mathcal{D}_{unlearn}$ (Figure 2) and reduces the success of the MIA (Figure 4 in the Appendix). Finally, Table 3 displays the results of both MIA methods. On the eICU dataset, R2D outperforms all other methods under the classic MIA and outperforms other certified methods under the MIA-U, and on the Lacuna-100 dataset, R2D is competitive under the classic MIA while outperforming all other methods under the MIA-U. Our results suggest that even with a small amount of perturbation, certified unlearning methods tend to defend against MIAs more successfully than non-certified

Table 2: Model performance (AUC) before and after unlearning. "Original Models" refers to the trained models before unlearning, and all other rows are unlearned models. We consider the relative decrease of the AUC on $\mathcal{D}_{unlearn}$ as an unlearning metric. The non-certified methods are applied to the R2D original model. We **bold** the two best results for each dataset.

| Algorithm | eICU | | | Lacuna-100 | | |
|---|---|---|---|---|---|---|
| | $\mathcal{D}_{retain}$ | $\mathcal{D}_{unlearn}$ | $\mathcal{D}_{test}$ | $\mathcal{D}_{retain}$ | $\mathcal{D}_{unlearn}$ | $\mathcal{D}_{test}$ |
| **Original Models** (no noise) | | | | | | |
| Hessian-Free | 0.7490 | 0.7614 | 0.7472 | 0.9851 | 0.9746 | 0.9737 |
| Constrained Newton Step | 0.7628 | 0.7860 | 0.7602 | 0.9983 | 0.9956 | 0.9856 |
| R2D | 0.7337 | 0.7451 | 0.7322 | 0.9989 | 0.9993 | 0.9844 |
| **Certified Methods** ($\sigma = 0.01$) | | | | | | |
| Hessian-Free | 0.7476 | **0.7587** (-0.0027) | 0.7465 | 0.9757 | 0.9555 (-0.0191) | 0.9652 |
| Constrained Newton Step | 0.7622 | 0.7848 (-0.0012) | 0.7601 | 0.9977 | 0.9949 (-0.0007) | 0.9847 |
| R2D 7-10% | 0.7335 | 0.7444 (-0.0007) | 0.7327 | 0.9606 | 0.9238 (-0.0755) | 0.9478 |
| R2D 36-37% | 0.7334 | 0.7441 (-0.0010) | 0.7327 | 0.9683 | **0.9269** (-0.0724) | 0.9525 |
| R2D 74% | 0.7333 | **0.7439** (-0.0012) | 0.7326 | 0.9712 | **0.8998** (-0.0995) | 0.9534 |
| **Noiseless Retrain** (R2D 100%) | 0.7335 | 0.7447 (-0.0004) | 0.7321 | 1.0000 | 0.9460 (-0.0533) | 0.9840 |
| **Non-certified Methods** | | | | | | |
| Finetune | 0.7352 | 0.7463 (+0.0012) | 0.7337 | 0.9997 | 0.9975 (-0.0018) | 0.9854 |
| Fisher Forgetting | 0.7310 | 0.7482 (+0.0031) | 0.7302 | 0.9982 | 0.9986 (-0.0007) | 0.9831 |
| SCRUB | 0.7336 | 0.7450 (-0.0001) | 0.7322 | 0.9989 | 0.9999 (+0.0006) | 0.9845 |

methods. Finally, Tables 6 and 7 in Appendix B.2 demonstrate that R2D is more computationally efficient than HF and CNS.

# 5 Conclusion

We propose R2D, the first black-box, first-order certified-unlearning algorithm for nonconvex functions, addressing theoretical and practical limitations of prior work. Our algorithm outperforms existing certified and non-certified methods in storage, computation, accuracy, and unlearning.

Table 3: Membership inference attack success (AUC). We **bold** the two best results for each dataset and attack method.

| Algorithm | eICU | | Lacuna-100 | |
|---|---|---|---|---|
| | **Classic MIA** | **MIA-U** | **Classic MIA** | **MIA-U** |
| **Certified** | | | | |
| ($\sigma = 0.01$) | | | | |
| HF | 0.5078 | 0.5108 | **0.4950** | 0.8460 |
| CNS | 0.5145 | 0.5144 | 0.6020 | 0.8480 |
| R2D 7-10% | 0.5047 | 0.5001 | 0.5625 | 0.7721 |
| R2D 36-37% | **0.5046** | **0.5001** | 0.5197 | **0.6779** |
| R2D 74% | **0.5044** | **0.4997** | 0.5333 | **0.7206** |
| **Noiseless Retrain** | 0.5060 | 0.5064 | **0.4974** | 0.8101 |
| **Non-certified** | | | | |
| Finetune | 0.5063 | 0.5066 | 0.6013 | 0.8497 |
| Fisher Forgetting | 0.5114 | 0.5102 | 0.5814 | 0.8735 |
| SCRUB | 0.5060 | 0.5056 | 0.6179 | 0.8586 |

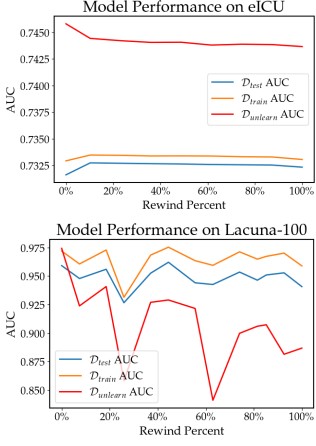

Figure 2: Model performance vs. rewinding for $\sigma = 0.01$.

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

# A   Proofs

The proof of Theorem 3.1 is provided in Appendix A.2. The proof of Corollary 3.2 is in Appendix A.2.2.

## A.1   Proof of Lemma 2.1

*Proof.* We have the following lemma from [13].

**Lemma A.1.** *[13] If $f(\theta)$ is continuously differentiable with L-Lipschitz gradient, $-f(\theta) + \frac{L}{2}||\theta||^2$ is convex.*

Now we define $\theta^*$ as the solution to the proximal problem as follows

$$\theta^* = prox_{-f,\eta}(\theta_{t+1}) = \arg\min_x\{-f(x) + \frac{1}{2\eta}||x - \theta_{t+1}||^2\},$$

which is well-defined due to Lemma A.1 and the fact that $\eta < \frac{1}{L}$, leading to strong convexity. Then the gradient of the objective function is zero at $\theta^*$ and thus

$$-\nabla f(\theta^*) + \frac{1}{\eta}(\theta^* - \theta_{t+1}) = 0,$$
$$\theta^* = \theta_{t+1} + \eta\nabla f(\theta^*).$$

$\square$

## A.2   Proof of Theorem 3.1

Like prior work in differential privacy and machine unlearning, our work hinges on the Gaussian mechanism for differential privacy, which ensures $(\epsilon, \delta)$-indistinguishability for normal random variables with the same variance.

**Theorem A.2.** *[14] Let $X \sim \mathcal{N}(\mu, \sigma^2\mathbb{I}_d)$ and $Y \sim \mathcal{N}(\mu', \sigma^2\mathbb{I}_d)$. Suppose $||\mu - \mu'||_2 \leq \Delta$. Then for any $\delta > 0$, $X$ and $Y$ are $(\epsilon, \delta)$-indistinguishable if*

$$\sigma = \frac{\Delta}{\epsilon}\sqrt{2\log(1.25/\delta)}.$$

Therefore, to prove Theorem 3.1, we need to bound the distance between the output of the unlearning algorithm and the learning algorithm. We can then add a sufficient amount of noise, scaled by this distance, to achieve $(\epsilon, \delta)$ unlearning.

*Proof.* Let $Z$ be a dataset of $m$ data points we would like to unlearn, where $m < n$. Let $\mathcal{D}$ represent the original full dataset and $\mathcal{D}' = \mathcal{D}\backslash Z$. Without loss of generality, define

$$f_{\mathcal{D}}(\theta) = \frac{1}{n}\sum_{i=1}^{n} f_{z_i}(\theta),$$

$$f_{\mathcal{D}'}(\theta) = \frac{1}{n-m}\sum_{i=1}^{n-m} f_{z_i}(\theta),$$

such that we have

$$f_{\mathcal{D}} = \frac{n-m}{n}f_{\mathcal{D}'}(\theta) + \frac{1}{n}\sum_{i=n-m+1}^{n} f_{z_i}(\theta),$$

$$f_{\mathcal{D}'} = \frac{n}{n-m}(f_{\mathcal{D}}(\theta) - \frac{1}{n}\sum_{i=n-m+1}^{n} f_{z_i}(\theta)).$$

Let $\{\theta_t\}_{t=0}^{T}$ represent the gradient descent iterates of the learning algorithm on $f_{\mathcal{D}}$, starting from $\theta_0$, and let $\{\theta_t'\}_{t=0}^{T}$ be the iterates of the learning algorithm on $f_{\mathcal{D}'}$ starting from the same $\theta_0$. Then we have

$$\theta_0 = \theta_0', \tag{7}$$
$$\theta_t = \theta_{t-1} - \eta\nabla f_{\mathcal{D}}(\theta_{t-1}), \tag{8}$$
$$\theta_t' = \theta_{t-1}' - \eta\nabla f_{\mathcal{D}'}(\theta_{t-1}'). \tag{9}$$

Let $\{\theta_t''\}_{t=0}^K$ represent the gradient descent iterates of the unlearning algorithm starting at $\theta_0'' = \theta_{T-K}$. Finally, let $\tilde{\theta} = \theta_K'' + \xi$ denote the iterate with Gaussian noise added.

We first bound the distance between $\theta_t$ and $\theta_t'$ as follows.

**Lemma A.3.** *Let $\{\theta_t\}_{t=0}^T$, $\{\theta_t'\}_{t=0}^T$ be defined as in (7). Then*

$$\|\theta_t - \theta_t'\| \leq \frac{2Gm}{Ln}[(1 + \frac{\eta Ln}{n-m})^t - 1].$$

*Proof.* We have

$$\nabla f_{\mathcal{D}'}(\theta) = \frac{n}{n-m}(\nabla f_{\mathcal{D}}(\theta) - \frac{1}{n}\sum_{i=n-m+1}^n \nabla f_{z_i}(\theta)),$$

$$\theta_t' = \theta_{t-1}' - \eta\frac{n}{n-m}(\nabla f_{\mathcal{D}}(\theta_{t-1}') - \frac{1}{n}\sum_{i=n-m+1}^n \nabla f_{z_i}(\theta_{t-1}')).$$

So we have for $\Delta_t = \theta_t - \theta_t'$,

$$\Delta_t = \Delta_{t-1} - \eta\nabla f_{\mathcal{D}}(\theta_{t-1}) + \eta\frac{n}{n-m}(\nabla f_{\mathcal{D}}(\theta_{t-1}') - \frac{1}{n}\sum_{i=n-m+1}^n \nabla f_{z_i}(\theta_{t-1}')),$$

$$= \Delta_{t-1} - \eta\nabla f_{\mathcal{D}}(\theta_{t-1}) + \eta\frac{n}{n-m}\nabla f_{\mathcal{D}}(\theta_{t-1}') - \eta\frac{1}{n-m}\sum_{i=n-m+1}^n \nabla f_{z_i}(\theta_{t-1}'),$$

$$= \Delta_{t-1} - \eta\nabla f_{\mathcal{D}}(\theta_{t-1}) + \eta\frac{n}{n-m}\nabla f_{\mathcal{D}}(\theta_{t-1}) + \eta\frac{n}{n-m}(\nabla f_{\mathcal{D}}(\theta_{t-1}') - \nabla f_{\mathcal{D}}(\theta_{t-1}))$$

$$- \eta\frac{1}{n-m}\sum_{i=n-m+1}^n \nabla f_{z_i}(\theta_{t-1}'),$$

$$= \Delta_{t-1} + \eta\frac{m}{n-m}\nabla f_{\mathcal{D}}(\theta_{t-1}) + \eta\frac{n}{n-m}(\nabla f_{\mathcal{D}}(\theta_{t-1}') - \nabla f_{\mathcal{D}}(\theta_{t-1})) - \eta\frac{1}{n-m}\sum_{i=n-m+1}^n \nabla f_{z_i}(\theta_{t-1}').$$

After taking the absolute value of each side, we obtain by the triangle inequality

$$\|\Delta_t\| \leq \|\Delta_{t-1}\| + \eta\frac{m}{n-m}\|\nabla f_{\mathcal{D}}(\theta_{t-1})\| + \eta\frac{n}{n-m}\|\nabla f_{\mathcal{D}}(\theta_{t-1}') - \nabla f_{\mathcal{D}}(\theta_{t-1})\| + \eta\frac{1}{n-m}\sum_{i=n-m+1}^n \|\nabla f_{z_i}(\theta_{t-1}')\|.$$

Since the gradient on each data sample is bounded by $G$, we have

$$\|\Delta_t\| \leq \|\Delta_{t-1}\| + \eta\frac{m}{n-m}G + \eta\frac{n}{n-m}\|\nabla f_{\mathcal{D}}(\theta_{t-1}') - \nabla f_{\mathcal{D}}(\theta_{t-1})\| - \eta\frac{1}{n-m}\sum_{i=n-m+1}^n G,$$

$$\|\Delta_t\| \leq \|\Delta_{t-1}\| + \eta\frac{n}{n-m}\|\nabla f_{\mathcal{D}}(\theta_{t-1}') - \nabla f_{\mathcal{D}}(\theta_{t-1})\| + \frac{2\eta Gm}{n-m}.$$

By Lipschitz smoothness of the gradient, we have

$$\|\Delta_t\| \leq \|\Delta_{t-1}\| + \eta\frac{n}{n-m}L\|\theta_{t-1}' - \theta_{t-1}\| + \frac{2\eta Gm}{n-m},$$

$$\|\Delta_t\| \leq \|\Delta_{t-1}\|(1 + \frac{\eta Ln}{n-m}) + \frac{2\eta Gm}{n-m}.$$

Since we have $\|\Delta_0\| = 0$, evaluating this recursive relationship yields for $t > 0$

$$\|\Delta_t\| \leq \frac{2\eta Gm}{n-m}\sum_{i=0}^{t-1}(1 + \frac{\eta Ln}{n-m})^i$$

$$\|\Delta_t\| \leq \frac{2\eta Gm}{n-m}\frac{(1 + \frac{\eta Ln}{n-m})^t - 1}{\frac{\eta Ln}{n-m}}$$

$$||\Delta_t|| \leq 2Gm \frac{(1 + \frac{\eta L n}{n-m})^t - 1}{Ln}$$

This bound takes advantage of the difference between $f_\mathcal{D}$ and $f_{\mathcal{D}'}$ as it decreases with large $n$, but it also grows exponentially with the number of iterates.

$\square$

**Lemma A.4.** *Let* $\{\theta''_t\}_{t=0}^K$ *represent the gradient descent iterates on* $f_{\mathcal{D}'}$ *starting at* $\theta''_0 = \theta_{T-K}$ *such that*

$$\theta''_t = \theta''_{t-1} - \eta \nabla f_{\mathcal{D}'}(\theta''_{t-1})$$

*Then*

$$||\theta'_T - \theta''_K|| \leq ||\theta_{T-K} - \theta'_{T-K}||(1 + \eta L)^K$$

*Proof.* Let $\Delta'_t = \theta'_{T-K+t} - \theta''_t$ such that $\Delta'_0 = ||\theta'_{T-K} - \theta''_0||$ and we bound it as follows

$$\Delta'_t = \theta'_{T-K+t} - \theta''_t = \theta'_{T-K+t-1} - \eta \nabla f_{\mathcal{D}'}(\theta'_{T-K+t-1}) - \theta''_{t-1} + \eta \nabla f_{\mathcal{D}'}(\theta''_{t-1})$$

$$= \Delta'_{t-1} - \eta \nabla f_{\mathcal{D}'}(\theta'_{T-K+t-1}) + \eta \nabla f_{\mathcal{D}'}(\theta''_{t-1})$$

$$||\Delta'_t|| \leq ||\Delta'_{t-1}|| + \eta ||\nabla f_{\mathcal{D}'}(\theta'_{T-K+t-1}) - \nabla f_{\mathcal{D}'}(\theta''_{t-1})||$$

By Lipschitz smoothness

$$||\Delta'_t|| \leq ||\Delta'_{t-1}|| + L\eta ||\theta'_{T-K+t-1} - \theta''_{t-1}||$$

$$||\Delta'_t|| \leq (1 + \eta L)||\Delta'_{t-1}|| \leq ||\Delta'_0||(1 + \eta L)^t$$

$\square$

Returning to the algorithm, suppose the learning algorithm has $T$ iterations and we backtrack for $K$ iterations. Then the difference between the output of the learning algorithm (without noise) on $f_{\mathcal{D}'}$ and the unlearning algorithm would be

$$||\theta'_T - \theta''_K|| \leq ||\Delta_{T-K}||(1 + \eta L)^K \leq \frac{2mG}{Ln}((1 + \frac{\eta L n}{n-m})^{T-K} - 1)(1 + \eta L)^K$$

where the bound on the right hand side decreases monotonically as $K$ increases from $0$ to $T$, as shown by the following. Let

$$h(K) = ((1 + \frac{\eta L n}{n-m})^{T-K} - 1)(1 + \eta L)^K.$$

Then the derivative is

$$h'(K) = (1 + \eta L)^K[((1 + \frac{\eta L n}{n-m})^{T-K} - 1)\log(1 + \eta L) - (1 + \frac{\eta L n}{n-m})^{T-K}\log(1 + \frac{\eta L n}{n-m})],$$

we observe that $h'(K) < 0$ for $K \in [0, T]$. Therefore $h(K)$ is decreasing.

Therefore by Theorem A.2, to achieve $\epsilon, \delta$-unlearning, we need to set the value of $\sigma$ as follows

$$\sigma = \frac{||\theta'_T - \theta''_K||\sqrt{2\log(1.25/\delta)}}{\epsilon} = \frac{2mG \cdot h(K)\sqrt{2\log(1.25/\delta)}}{Ln\epsilon}.$$

Now we prove the utility guarantee. For nonconvex smooth functions, we know by standard analysis that for the gradient descent iterates $\theta''_t$, we have

$$\frac{\eta}{2} \sum_{t=0}^K ||\nabla f_{\mathcal{D}'}(\theta''_t)|| \leq \sum_{t=0}^K f_{\mathcal{D}'}(\theta''_t) - f_{\mathcal{D}'}(\theta''_{t+1}) = f_{\mathcal{D}'}(\theta''_0) - f_{\mathcal{D}'}(\theta''_K)$$

Now we consider the progress of the iterates $\theta_t$ on $f_{\mathcal{D}'}(\theta)$. By Lipschitz smoothness, we have

$$f_{\mathcal{D}'}(\theta_{t+1}) \leq f_{\mathcal{D}'}(\theta_t) + \langle \nabla f_{\mathcal{D}'}(\theta_t), -\eta \nabla f_{\mathcal{D}}(\theta_t) \rangle + \frac{L}{2}||\eta \nabla f_{\mathcal{D}}(\theta_t)||^2$$

We have

$$\nabla f_{\mathcal{D}}(\theta_t) = \frac{n-m}{n}\nabla f_{\mathcal{D}'}(\theta_t) + \frac{1}{n}\sum_{i=n-m+1}^{n}\nabla f_{z_i}(\theta_t)$$

$$f_{\mathcal{D}'}(\theta_{t+1}) \leq f_{\mathcal{D}'}(\theta_t) - \eta\langle\nabla f_{\mathcal{D}'}(\theta_t), \frac{n-m}{n}\nabla f_{\mathcal{D}'}(\theta_t) + \frac{1}{n}\sum_{i=n-m+1}^{n}\nabla f_{z_i}(\theta_t)\rangle$$

$$+ \frac{L\eta^2}{2}||\frac{n-m}{n}\nabla f_{\mathcal{D}'}(\theta_t) + \frac{1}{n}\sum_{i=n-m+1}^{n}\nabla f_{z_i}(\theta_t)||^2$$

$$\leq f_{\mathcal{D}'}(\theta_t) - \eta\frac{n-m}{n}||\nabla f_{\mathcal{D}'}(\theta_t)||^2 - \eta\langle\nabla f_{\mathcal{D}'}(\theta_t), \frac{1}{n}\sum_{i=n-m+1}^{n}\nabla f_{z_i}(\theta_t)\rangle$$

$$+ \frac{L\eta^2}{2}||\frac{n-m}{n}\nabla f_{\mathcal{D}'}(\theta_t) + \frac{1}{n}\sum_{i=n-m+1}^{n}\nabla f_{z_i}(\theta_t)||^2$$

$$\leq f_{\mathcal{D}'}(\theta_t) - \eta\frac{n-m}{n}||\nabla f_{\mathcal{D}'}(\theta_t)||^2 - \eta\langle\nabla f_{\mathcal{D}'}(\theta_t), \frac{1}{n}\sum_{i=n-m+1}^{n}\nabla f_{z_i}(\theta_t)\rangle$$

$$+ L\eta^2||\frac{n-m}{n}\nabla f_{\mathcal{D}'}(\theta_t)||^2 + L\eta^2||\frac{1}{n}\sum_{i=n-m+1}^{n}\nabla f_{z_i}(\theta_t)||^2$$

$$\leq f_{\mathcal{D}'}(\theta_t) - \eta\frac{n-m}{n}(1 - \frac{L\eta(n-m)}{n})||\nabla f_{\mathcal{D}'}(\theta_t)||^2 - \eta\langle\nabla f_{\mathcal{D}'}(\theta_t), \frac{1}{n}\sum_{i=n-m+1}^{n}\nabla f_{z_i}(\theta_t)\rangle$$

$$+ L\eta^2\frac{1}{n}\sum_{i=n-m+1}^{n}||\nabla f_{z_i}(\theta_t)||^2$$

$$\leq f_{\mathcal{D}'}(\theta_t) - \eta\frac{n-m}{n}(1 - \frac{L\eta(n-m)}{n})||\nabla f_{\mathcal{D}'}(\theta_t)||^2 + \eta\frac{G^2m}{n} + L\eta^2\frac{mG}{n}$$

Let the step size $\eta$ be bounded such that $\eta \leq \frac{n}{2(n-m)L}$, then we have

$$f_{\mathcal{D}'}(\theta_{t+1}) \leq f_{\mathcal{D}'}(\theta_t) - \frac{\eta(n-m)}{2n}||\nabla f_{\mathcal{D}'}(\theta_t)||^2 + \frac{\eta G^2m}{n} + \frac{L\eta^2Gm}{n}$$

Rearrange the terms to get

$$\frac{\eta(n-m)}{2n}||\nabla f_{\mathcal{D}'}(\theta_t)||^2 \leq f_{\mathcal{D}'}(\theta_t) - f_{\mathcal{D}'}(\theta_{t+1}) + \frac{\eta G^2m}{n} + \frac{L\eta^2Gm}{n}$$

Sum from $t=0$ to $T-K-1$

$$\frac{\eta(n-m)}{2n}\sum_{t=0}^{T-K-1}||\nabla f_{\mathcal{D}'}(\theta_t)||^2 \leq f_{\mathcal{D}'}(\theta_0) - f_{\mathcal{D}'}(\theta_{T-K}) + (T-K-1)(\frac{\eta G^2m}{n} + \frac{L\eta^2Gm}{n})$$

$$\sum_{t=0}^{T-K-1}||\nabla f_{\mathcal{D}'}(\theta_t)||^2 \leq \frac{2n}{\eta(n-m)}(f_{\mathcal{D}'}(\theta_0) - f_{\mathcal{D}'}(\theta_{T-K}) + (T-K-1)(\frac{\eta G^2m}{n} + \frac{L\eta^2Gm}{n}))$$

$$\sum_{t=0}^{T-K-1}||\nabla f_{\mathcal{D}'}(\theta_t)||^2 \leq \frac{2n}{\eta(n-m)}(f_{\mathcal{D}'}(\theta_0) - f_{\mathcal{D}'}(\theta_{T-K})) + (T-K-1)(\frac{2G^2m}{n-m} + \frac{2L\eta Gm}{n-m})$$

From standard analysis of gradient descent on nonconvex functions, we know

$$\eta\sum_{t=0}^{K}||\nabla f_{\mathcal{D}'}(\theta_t'')||^2 \leq f_{\mathcal{D}'}(\theta_0'') - f_{\mathcal{D}'}(\theta_K'')$$

$$\sum_{t=0}^{K}||\nabla f_{\mathcal{D}'}(\theta_t'')||^2 \leq \frac{1}{\eta}(f_{\mathcal{D}'}(\theta_0'') - f_{\mathcal{D}'}(\theta_K''))$$

Summing the equations yields

$$\sum_{t=0}^{T-K-1} ||\nabla f_{\mathcal{D}'}(\theta_t)||^2 + \sum_{t=0}^{K} ||\nabla f_{\mathcal{D}'}(\theta_t'')||^2$$

$$\leq \frac{2n}{\eta(n-m)}(f_{\mathcal{D}'}(\theta_0) - f_{\mathcal{D}'}(\theta_{T-K})) + (T-K-1)(\frac{2G^2m}{n-m} + \frac{2L\eta Gm}{n-m}) + \frac{1}{\eta}(f_{\mathcal{D}'}(\theta_0'') - f_{\mathcal{D}'}(\theta_K''))$$

$$\leq \frac{2n}{\eta(n-m)}(f_{\mathcal{D}'}(\theta_0) - f_{\mathcal{D}'}(\theta_K'')) + (T-K-1)(\frac{2G^2m}{n-m} + \frac{2L\eta Gm}{n-m})$$

We can expand $\mathbb{E}[||\nabla f_{\mathcal{D}'}(\tilde{\theta}'')||^2]$ as follows

$$||\nabla f_{\mathcal{D}'}(\tilde{\theta}'')||^2 = ||\nabla f_{\mathcal{D}'}(\tilde{\theta}'')||^2 + \mathbb{E}[2\nabla f_{\mathcal{D}'}(\tilde{\theta}'')^T \xi] + \mathbb{E}[||\xi||^2]$$
$$= ||\nabla f_{\mathcal{D}'}(\tilde{\theta}'')||^2 + \sigma^2.$$

So we have

$$\frac{1}{T}\Big[\sum_{t=0}^{T-K-1} ||\nabla f_{\mathcal{D}'}(\theta_t)||^2 + \sum_{t=0}^{K-1} ||\nabla f_{\mathcal{D}'}(\theta_t'')||^2 + \mathbb{E}[||\nabla f_{\mathcal{D}'}(\tilde{\theta}'')||^2]\Big] \leq \sigma^2 + \frac{2n(f_{\mathcal{D}'}(\theta_0) - f_{\mathcal{D}'}(\theta_K''))}{T\eta(n-m)}$$

$$+ \frac{T-K-1}{T}(\frac{2G^2m}{n-m} + \frac{2L\eta Gm}{n-m})$$

### A.2.1 Proof of Corollary 3.3

To prove Corollary 3.3, we first need to obtain empirical risk bounds for PL functions. Corollary A.5 states that the empirical risk converges linearly with both $T$ and $K$.

**Corollary A.5.** *Suppose the conditions of Theorem 3.1 hold and in addition, $f_{\mathcal{D}'}$ satisfies the PL condition with parameter $\mu$. Let $f_{\mathcal{D}'}^*$ represent the global optimal value of $f_{\mathcal{D}'}$. Then*

$$\mathbb{E}[f_{\mathcal{D}'}(\tilde{\theta}'')] - f_{\mathcal{D}'}^* \leq L\sqrt{d}\sigma + (1-\eta\mu)^K \frac{G^2m + L\eta Gm}{\mu(n-m)} + (1 - \frac{\eta\mu(n-m)}{n})^{T-K}(1-\eta\mu)^K (f_{\mathcal{D}'}(\theta_0) - f_{\mathcal{D}'}^*)$$

*where $\sigma$ is defined in (3) and the expectation is taken with respect to the Gaussian noise added at the end of $U$.*

*Proof.* As before, we first consider the gradient descent iterates on $f_{\mathcal{D}'}$. For $\mu$-PL and smooth functions, we know that for the iterates $\theta_t''$, we have [27]

$$f_{\mathcal{D}'}(\theta_t'') - f_{\mathcal{D}'}^* \leq (1-\eta\mu)^t(f_{\mathcal{D}'}(\theta_0'') - f_{\mathcal{D}'}^*).$$

Now we track the progress of the iterates $\theta_t$ on $f_{\mathcal{D}'}$. By Lipschitz smoothness and the above analysis, we have

$$f_{\mathcal{D}'}(\theta_{t+1}) - f_{\mathcal{D}'}(\theta_t) \leq \langle \nabla f_{\mathcal{D}'}(\theta_t), -\eta \nabla f_{\mathcal{D}}(\theta_t)\rangle + \frac{L}{2}||\eta \nabla f_{\mathcal{D}}(\theta_t)||^2,$$

$$f_{\mathcal{D}'}(\theta_{t+1}) - f_{\mathcal{D}'}(\theta_t) \leq -\eta \frac{n-m}{n}(1 - \frac{L\eta(n-m)}{n})||\nabla f_{\mathcal{D}'}(\theta_t)||^2 + \eta\frac{G^2m}{n} + L\eta^2\frac{mG}{n}.$$

Let the step size $\eta$ be bounded such that $\eta \leq \frac{n}{2(n-m)L}$, then we have

$$f_{\mathcal{D}'}(\theta_{t+1}) - f_{\mathcal{D}'}(\theta_t) \leq -\frac{\eta(n-m)}{2n}||\nabla f_{\mathcal{D}'}(\theta_t)||^2 + \frac{\eta G^2m}{n} + \frac{L\eta^2 Gm}{n}.$$

By the PL inequality, we have

$$f_{\mathcal{D}'}(\theta_{t+1}) - f_{\mathcal{D}'}(\theta_t) \leq -\frac{\eta\mu(n-m)}{n}(f_{\mathcal{D}'}(\theta_t) - f_{\mathcal{D}'}^*) + \frac{\eta G^2m}{n} + \frac{L\eta^2 Gm}{n},$$

$$f_{\mathcal{D}'}(\theta_{t+1}) - f_{\mathcal{D}'}^* \leq (1 - \frac{\eta\mu(n-m)}{n})(f_{\mathcal{D}'}(\theta_t) - f_{\mathcal{D}'}^*) + \frac{\eta G^2m}{n} + \frac{L\eta^2 Gm}{n}.$$

We evaluate this recursive relationship to obtain

$$f_{\mathcal{D}'}(\theta_{t+1}) - f_{\mathcal{D}'}^* \le (1 - \frac{\eta\mu(n-m)}{n})^{t+1}(f_{\mathcal{D}'}(\theta_0) - f_{\mathcal{D}'}^*) + \frac{1}{n}(\eta G^2 m + L\eta^2 Gm)\sum_{i=0}^{t}(1 - \frac{\eta\mu(n-m)}{n})^i$$

$$f_{\mathcal{D}'}(\theta_{t+1}) - f_{\mathcal{D}'}^* \le (1 - \frac{\eta\mu(n-m)}{n})^{t+1}(f_{\mathcal{D}'}(\theta_0) - f_{\mathcal{D}'}^*) + \frac{1}{n}(\eta G^2 m + L\eta^2 Gm)\sum_{i=0}^{t}(1 - \frac{\eta\mu(n-m)}{n})^i$$

$$\le (1 - \frac{\eta\mu(n-m)}{n})^{t+1}(f_{\mathcal{D}'}(\theta_0) - f_{\mathcal{D}'}^*) + \frac{1}{n}(\eta G^2 m + L\eta^2 Gm)\frac{n}{\eta\mu(n-m)}$$

$$f_{\mathcal{D}'}(\theta_{t+1}) - f_{\mathcal{D}'}^* \le (1 - \frac{\eta\mu(n-m)}{n})^{t+1}(f_{\mathcal{D}'}(\theta_0) - f_{\mathcal{D}'}^*) + \frac{G^2 m + L\eta Gm}{\mu(n-m)}$$

$$f_{\mathcal{D}'}(\theta_0'') - f_{\mathcal{D}'}^* = f_{\mathcal{D}'}(\theta_{T-K}) - f_{\mathcal{D}'}^* \le (1 - \frac{\eta\mu(n-m)}{n})^{T-K}(f_{\mathcal{D}'}(\theta_0) - f_{\mathcal{D}'}^*) + \frac{G^2 m + L\eta Gm}{\mu(n-m)}$$

$$f_{\mathcal{D}'}(\theta_K'') - f_{\mathcal{D}'}^* \le (1 - \frac{\eta\mu(n-m)}{n})^{T-K}(1 - \eta\mu)^K(f_{\mathcal{D}'}(\theta_0) - f_{\mathcal{D}'}^*) + (1 - \eta\mu)^K\frac{G^2 m + L\eta Gm}{\mu(n-m)} \tag{10}$$

By Lipschitz smoothness, we have

$$\begin{aligned}
f_{\mathcal{D}'}(\tilde{\theta}'') - f_{\mathcal{D}'}^* \le{}& f_{\mathcal{D}'}(\tilde{\theta}'') - f_{\mathcal{D}'}(\theta_K'') + (1 - \frac{\eta\mu(n-m)}{n})^{T-K}(1 - \eta\mu)^K(f_{\mathcal{D}'}(\theta_0) - f_{\mathcal{D}'}^*) \\
&+ (1 - \eta\mu)^K\frac{G^2 m + L\eta Gm}{\mu(n-m)} \\
\le{}& L\|\tilde{\theta}'' - \theta_K''\| + (1 - \frac{\eta\mu(n-m)}{n})^{T-K}(1 - \eta\mu)^K(f_{\mathcal{D}'}(\theta_0) - f_{\mathcal{D}'}^*) \\
&+ (1 - \eta\mu)^K\frac{G^2 m + L\eta Gm}{\mu(n-m)} \\
={}& L\|\xi\| + (1 - \frac{\eta\mu(n-m)}{n})^{T-K}(1 - \eta\mu)^K(f_{\mathcal{D}'}(\theta_0) - f_{\mathcal{D}'}^*) \\
&+ (1 - \eta\mu)^K\frac{G^2 m + L\eta Gm}{\mu(n-m)}
\end{aligned} \tag{11}$$

If we take the expectation on both sides with respect to the noise added at the end of the algorithm $U$, we have

$$\mathbb{E}[f_{\mathcal{D}'}(\tilde{\theta}'')] - f_{\mathcal{D}'}^* \le L\sqrt{d}\sigma + (1 - \frac{\eta\mu(n-m)}{n})^{T-K}(1 - \eta\mu)^K(f_{\mathcal{D}'}(\theta_0) - f_{\mathcal{D}'}^*) + (1 - \eta\mu)^K\frac{G^2 m + L\eta Gm}{\mu(n-m)}$$

$\square$

Now we can prove Corollary 3.3. Bounds on generalization can be derived using seminal results in algorithmic stability [16]. Prior work on the generalization ability of unlearning algorithms focus on the strongly convex case [38, 37, 31, 45], which feature excess risk bounds for the empirical risk minimizer when the loss is strongly convex [39]. By considering PL functions, we are, to the best of our knowledge, the first to consider unlearning generalization in a nonconvex setting. In contrast, algorithms on PL objective functions can at best satisfy pointwise [7] or on-average [29] stability. The following result derives from Theorem 1 of [29]. We utilize the fact that for $w^* \in \arg\min_{\theta\in\Theta} F(\theta)$, we have $\mathbb{E}[f_{\mathcal{D}'}^*] \le \mathbb{E}[f_{\mathcal{D}'}(w^*)] = F(w^*) = F^*$, and we also slightly modify the result to include the stronger bounded gradient assumption used in our work.

**Lemma A.6.** *[29] For $F$ defined as $F(\theta) = \mathbb{E}_{z\sim\mathcal{Z}}[f_z(\theta)]$*

*and $\theta$ as the output of an algorithm dependent on $\mathcal{D}'$, we have*

$$\mathbb{E}[F(\theta) - F^*] \le \frac{2G^2}{(n-m)\mu} + \frac{L}{2\mu}\mathbb{E}[f_{\mathcal{D}'}(\theta) - f_{\mathcal{D}'}^*].$$

We obtain the result by substituting in (10). The population risk can be decomposed to three terms: an $O(1/n)$ term representing the difference with empirical risk, an $O(\sqrt{d}\sigma)$ term representing the Gaussian noise, and a term representing the convergence of the underlying optimization framework. The first two terms are roughly equivalent across most works because they follow from fundamental principles.

### A.2.2  Proof of Corollary 3.2

We want to determine the value of $K$ required to maintain a privacy level $\epsilon$ for a chosen level of noise $\sigma$. From (3) we have

$$\epsilon = \frac{h(K)2mG\sqrt{2\log(1.25/\delta)}}{\sigma Ln}$$

Although this does not have an exact explicit solution for $K(\epsilon)$, we can bound $h(K)$ as follows

$$h(K) \leq ((1+\frac{\eta Ln}{n-m})^{T-K}-1)(1+\frac{\eta Ln}{n-m})^{K} = (1+\frac{\eta Ln}{n-m})^{T}-(1+\frac{\eta Ln}{n-m})^{K}$$

Then we have

$$\epsilon \leq \frac{2mG\sqrt{2\log(1.25/\delta)}}{\sigma Ln}((1+\frac{\eta Ln}{n-m})^{T}-(1+\frac{\eta Ln}{n-m})^{K})$$

$$(1+\frac{\eta Ln}{n-m})^{K} \leq (1+\frac{\eta Ln}{n-m})^{T}-\frac{\sigma Ln\epsilon}{2mG\sqrt{2\log(1.25/\delta)}}$$

$$K \leq \frac{\log\left((1+\frac{\eta Ln}{n-m})^{T}-\frac{\sigma Ln\epsilon}{2mG\sqrt{2\log(1.25/\delta)}}\right)}{\log(1+\frac{\eta Ln}{n-m})}$$

We therefore obtain a close upper bound on $K(\epsilon)$ for fixed $\sigma$. In practice we can choose $K$ equal to this bound to ensure the privacy guarantee is achieved. We show in Figure 3 that this bound is close to tight for $0 < \epsilon \leq 1$ and real-world parameters.

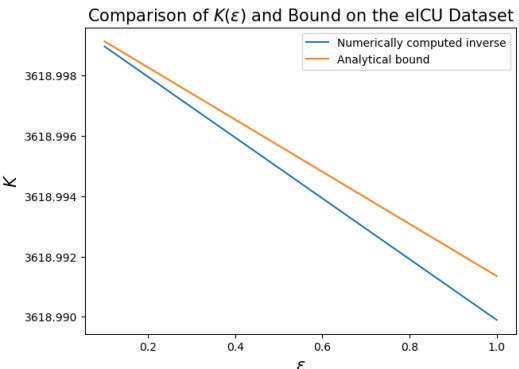

Figure 3: Comparison of $K(\epsilon)$ and analytically derived bound using real-world parameters from the eICU dataset.

## B  Experiments

In Appendix B.1, we review the details of our experimental implementations, including the membership inference attacks (Appendix B.1.1), and baseline methods (Appendix B.1.2). In Appendix B.2, we provide computation time experiments. In Appendix B.5, we present the full versions of Tables 2 and 3, including 1-sigma error bars for the MIA results, and tables for the noiseless version of the certified unlearning algorithms. We review the impact of rewinding on the attack success in Figure 4.

## B.1 Implementation Details

Code is open-sourced at the following GitHub link: https://github.com/siqiaomu/r2d.

**Datasets.** We consider two datasets: the eICU Collaborative Database, a large multi-center ICU database [36], and the Lacuna-100 dataset, a subset of the VGGFace2 [5] dataset, which we construct using the MAAD-Face annotations [44] and following the procedure in [20].

The eICU dataset can be obtained after following the instructions at https://eicu-crd.mit.edu/gettingstarted/access/. We combine the `patient`, `apacheApsVar`, `apachePredVar`, and `apachePatientResult` tables to form our dataset, predicting whether the hospital length of stay is longer or shorter than a week with the following predictive variables: `["age", "intubated", "vent", "dialysis", "eyes", "motor", "verbal", "meds", "urine", "wbc", "temperature", "respiratoryrate", "sodium", "heartrate", "meanbp", "ph", "hematocrit", "creatinine", "albumin", "pao2", "pco2", "bun", "glucose", "bilirubin", "fio2", "gender", "admitsource", "admitdiagnosis", "thrombolytics", "aids", "hepaticfailure", "lymphoma", "metastaticcancer", "leukemia", "immunosuppression", "cirrhosis", "diabetes"]`

Code for preprocessing the Lacuna-100 dataset is available at the GitHub repository linked above. The MAAD-Face annotations are available at https://github.com/pterhoer/MAAD-Face and the VGG-Face dataset is available at https://www.robots.ox.ac.uk/~vgg/data/vgg_face2/.

**Model architecture.** We perform experiments on a multilayer perceptron with three hidden layers and a ResNet-18 deep neural network. Because our analysis requires smooth functions, we replace all ReLU activations with SmeLU activations [40]. We also do not use Batch Normalization layers since we want to train with a constant step size. These steps may affect model performance but improve the soundness of our experiments by allowing estimation of the smoothness constant $L$.

**Gradient bound estimation.** To estimate the gradient bound $G$, we compute the norm of each mini-batch gradient at each step of the training process, and take the maximum of these values as the estimate.

**Lipschitz constant estimation.** To estimate the Lipschitz constant $L$, we perturb the model weights after training by Gaussian noise with $\sigma = 0.01$, and we estimate the Lipschitz constant by computing

$$\hat{L} = \frac{||\theta_1 - \theta_2||}{||\nabla f(\theta_1) - \nabla f(\theta_2)||}.$$

We sample $100$ perturbed weight samples and take the maximum of all the estimates $\hat{L}$ to be our Lipschitz constant.

Table 4: R2D Experiment parameters for the eICU and Lacuna-100 datasets.

| Experiment Parameter | eICU and MLP | Lacuna-100 and ResNet-18 |
|---|---|---|
| Size of training dataset $n$ | 94449 | 32000 |
| Number of users | 119282 | 100 |
| Percent data unlearned | $\sim 1\%$ | $\sim 2\%$ |
| Number of model parameters $d$ | 136386 | 11160258 |
| Batch size | 2048 | 512 |
| $L$ | 0.2065 | |
| $G$ | 0.5946 | |
| $\eta$ | 0.01 | 0.01 |
| Number of training epochs | 78 | 270 |

**Model selection.** To simulate real-world practices, we perform model selection during the initial training on the full dataset by training until the validation loss converges, and then selecting the model parameters with the lowest validation loss. We treat the selected iteration as the final training iterate.

**Unlearning metrics.** Many prior works solely use the error on the unlearned dataset as an unlearning metric, theorizing that the model should perform poorly on data it has never seen before. However, this is questionable in our context. For example, for Lacuna-100, unlearning a user's facial data does not preclude the model from correctly classifying their gender later, especially if the user's photos

are clear and easily identifiable. Moreover, since the unlearned data contains a small subset of the users in the training data, it is also likely to have lower variance. As a result, the unlearned model may even perform better on the unlearned data than on the training data, which is shown to be true for the eICU dataset. We therefore consider the performance on the unlearned data before and after unlearning (as shown in Figure 1g).

**Hardware.** All experiments were run using PyTorch 2.5.1 and CUDA 12.4, either on an Intel(R) Xeon(R) Silver 4214 CPU (2.20GHz) with an NVIDIA GeForce RTX 2080 (12 GB) or on an Intel(R) Xeon(R) Silver 4208 CPU (2.10GHz) with an NVIDIA RTX A6000 GPU (48 GB). Almost all experiments require less than 8 GB of GPU VRAM, except for running the HF algorithm on the Lacuna-100 dataset, which requires at least 29 GB of GPU VRAM to implement the scaled-down version.

**Additional details.** We use the same Gaussian noise vector for unlearned R2D under the same seed, rescaled to different standard deviations. We found that using unique noise for every hyperparameter combination resulted in similar but noisier trends. When initialized with the same seed, the `torch.randn` method outputs the same noise vector rescaled by the standard deviation. We use a different seed for the original model perturbations.

### B.1.1 Membership Inference Attacks

**Implementation.** We conduct MIAs to assess if information about the unlearned data is retained in the model weights and reflected in the model outputs. Prior works typically perform the attack comparing the unlearned dataset and the test dataset, the latter of which represents data previously unseen by the model. However, in our setting, the model may perform well on users present in both the training data and test data, while performing poorly on the users that are unlearned. It is therefore more appropriate to conduct the membership inference attack on out-of-distribution (OOD) data that contains data from *users* absent from the training data. For the Lacuna-100 dataset, which has fewer users and many samples per user, we construct an OOD dataset using an additional 100 users from the VGGFace2 database. For the eICU dataset, which has many users and a few samples per user, we construct the OOD dataset from data from the test set belonging to users not present in the training set.

We perform two different types of MIAs: the classic MIA approach based on [41], and an advanced attack tailored to the unlearning setting based on [9] (MIA-U). For both problem settings, we combine a subsample of the constructed OOD set with the unlearned set to form a 50-50 balanced training set for the attack model. We train a logistic regression model to perform binary classification, and we measure the success of the MIA by the attack model AUC on the data samples, computed via $k$-fold cross-validation ($k = 5$) repeated 10 times. For the classic MIA, we also repeat undersampling 100 times to stabilize the performance. We provide error bars for the MIA results in Appendix B.5.

The classic MIA considers the output of the unlearned model on the data. For the eICU setting, we consider the loss on the data, and for Lacuna-100, we consider the loss and logits on the data. The MIA-U leverages both the outputs of the unlearned and the original model, computing either a difference or a concatenation of the two posteriors to pass to the attack model. We found that using the Euclidean distance was most effective at preventing overfitting and improving the performance of the attack, which aligns with the observations of [9]. To study the effects of the MIA-U for certified unlearning, we add noise to both the original model and unlearned model before performing the attack. Both [48] and [37] address a weaker yet technically equivalent definition of certified unlearning, introduced in [38], that only considers indistinguishability with respect to the output of the unlearning algorithms, with or without certain data samples, as opposed to the definition used in our work, which considers indistinguishability with respect to the learning algorithm and the unlearning algorithm. As a result, neither [48] nor [37] require adding noise during the initial learning process, which may lead to a *more* successful MIA. However, their theoretical results naturally extend to the strong definition used in our paper, so for a fair comparison, we add noise to the original models of the other certified unlearning algorithms. Despite this, our algorithm still outperforms other certified unlearning algorithms in defending against the attacks.

**Additional Discussion of MIAs.** Although MIAs are a standard tool for testing user privacy and unlearning, we observe that they are an imperfect metric. MIAs are highly sensitive to distributional differences due to the non-uniform sampling of the unlearned data, and this sensitivity is stronger when the model performs well. This may explain why the MIA performs well even for the fully

retrained model, but is less successful on any of the certified methods that require adding uniform noise. It also explains why the attacks are weaker on the eICU dataset than on the Lacuna-100 dataset, since the model on the former dataset achieves a lower accuracy than on the latter.

On the flip side, MIAs can also react to distributional differences with deceptively good performance. For example, for very large $\sigma$, the perturbed model may perform poorly by classifying all data in a single class. If the class distribution in the unlearned dataset is different from the class distribution in the non-training dataset, the MIA may succeed by simply identifying the majority class in the unlearned dataset. To counter this effect, we ensure that the OOD dataset used for MIA and the unlearned dataset have the same class distribution. This nonlinear relationship is captured in the Classic MIA plot in Figure 1c, which shows that the Attack AUC is highest for very large or very small $\sigma$.

Ultimately, our results suggest that some caution is required when using MIAs to evaluate unlearning, especially in the real world where models may not be extremely well-performing or data may not be i.i.d. We find that the MIAs are generally less successful in our non-i.i.d. setting compared to the standard settings of other papers. A future direction of this research is developing a specialized MIA for the OOD setting presented in this paper.

### B.1.2 Baseline Methods

To implement the certified baseline methods, we use our estimated values of $L$ and $G$ when applicable, and we choose the step size and batch size so that the learning algorithms are sufficiently converged for our problems. For other parameters, such as minimum Hessian eigenvalue or Hessian Lipschitz constant, we use the values in the original works. We trained each unique learning algorithm on each dataset until convergence before adding noise. Tables 13 and 15 report the AUC of the noiseless trained models of the certified algorithms, showing that they achieve comparable performance prior to unlearning. Since HF requires $O(nd)$ storage (and $O(ndT)$ storage during training to compute the unlearning vectors), which is impractically large for our datasets and models, we follow their precedent and employ a scaled-down $O(md)$ version that only computes the vectors for the samples we plan to unlearn.

Table 5: Experiment parameters of HF and CNS for the eICU and Lacuna-100 datasets.

| | Experiment Parameter | eICU | Lacuna-100 |
|---|---|---|---|
| **Hessian-Free Unlearning** | | | |
| | Batch size | 512 | 256 |
| | $\eta_0$ | 0.1 | 0.1 |
| | Step size decay | 0.995 | 0.995 |
| | Gradient norm clipping | 5 | 5 |
| | Number of training epochs | 15 | 25 |
| | Optimizer | SGD | SGD |
| **Constrained Newton Step** | | | |
| | Batch size | 128 | 128 |
| | $\eta$ | 0.001 | 0.001 |
| | Weight decay | 0.0005 | 0.0005 |
| | Parameter norm constraint $R$ | 10 | 21 |
| | Number of training epochs | 30 | 30 |
| | Convex constant $\lambda$ | 200 | 2,000 |
| | Hessian scale $H$ | 50,000 | 50,000 |
| | Optimizer | Adam | Adam |

Table 5 shows the parameters used for implementing the certified unlearning baselines. Additional information can be found in the GitHub repository. The implementation of [48] is based on code from https://github.com/zhangbinchi/certified-deep-unlearning, and the implementation of [37] is based on code from https://github.com/XinbaoQiao/Hessian-Free-Certified-Unlearning.

The implementations of the non-certified baseline methods are also available in the GitHub repository. The hyperparameters for these methods are as follows:

- Finetune: 10 epochs

- Fisher Forgetting: alpha=1e-6 for eICU, alpha=1e-8
- SCRUB: See the following notebooks in the GitHub repository for full SCRUB hyperparameters, including `lacuna_baseline_implementation.ipynb` and `eicu_baseline_implementation.ipynb`.

## B.2 Computation Time Experiments

Because we desire machine unlearning algorithms that achieve a computational advantage over training from scratch, it is useful to compare the amount of time required for learning and unlearning for different algorithms. The results in Table 6 and 7 demonstrate that R2D unlearning is significantly more computationally efficient than HF and is competitive compared to CNS. These results were achieved on an NVIDIA RTX A6000 GPU (48 GB). As expected based on its straightforward algorithmic structure, the unlearning time of R2D is proportional to the number of rewind iterations.

Although these results depend on our implementation of the learning algorithms of HF and CNS, we observe that as shown in Table 5, we use fewer training iterations for HF compared to R2D. Despite this, HF requires *22-88 times more* compute time during learning, highlighting the benefits of black-box algorithms that dovetail with standard training practices and do not require complicated data-saving procedures. Similarly, CNS also requires fewer training iterations than R2D because it allows the use of advanced optimization techniques like momentum and regularization. However, because CNS unlearning requires an expensive second-order operation independent of the training process, it only displays a moderate computational advantage on the Lacuna-100 dataset and no advantage on the eICU dataset. In contrast, by construction R2D unlearning will *always* require less computation than learning, demonstrating the benefits of a cheap first-order approach.

Table 6: Comparison of computation time on the eICU dataset, with $23\%$ or $49\%$ of training iterations for R2D.

| Algorithm | Learning Time | Unlearning Time |
|---|---|---|
| Rewind-to-Delete (R2D), 23% | 230.76 sec | 58.48 sec |
| Rewind-to-Delete (R2D), 49% | 230.76 sec | 113.69 sec |
| Hessian-Free (HF) | 5.65 hours | 0.01 sec |
| Constrained Newton Step (CNS) | 266.0 sec | 293.6 sec |

Table 7: Comparison of computation time on the Lacuna-100 dataset, with $19\%$ or $44\%$ of training iterations for R2D.

| Algorithm | Learning Time | Unlearning Time |
|---|---|---|
| Rewind-to-Delete (R2D), 19% | 4.57 hours | 0.91 hours |
| Rewind-to-Delete (R2D), 44% | 4.57 hours | 2.06 hours |
| Hessian-Free (HF) | 4.23 days | 0.00 hours |
| Constrained Newton Step (CNS) | 0.58 hours | 0.32 hours |

As for storage, we note that even with the stripped-down version of HF, it still requires 1.5 GB of storage for the eICU dataset and 136GB for the Lacuna-100 dataset during training, whereas R2D only needs to store the model weights at a checkpoint and the final iterate $(2 \times 8)$ MB for eICU and $(2 \times 45$ MB for Lacuna-100).

## B.3 Proximal Point Experiments

We implement the proximal point method (Algorithm 3) and evaluate its ability to reconstruct prior checkpoints. Our results are shown in Table 8. We measure the success by the Euclidean distance between the original checkpoint and the reconstructed checkpoint, observing that they are close for small $K$ but grow apart as we rewind past half the original training trajectory, due to compounding approximation errors that ultimately leads to instability in convergence. Taking into account the model dimension, our results show that we can quite faithfully reconstruct checkpoints for moderate $K$.

We also evaluated the computational cost of Algorithm 3 and found that in most cases, reconstructing the checkpoint has a minor impact on the overall runtime. In Tables 9 and 10, we include the time it takes to reconstruct the checkpoint into the learning time. For eICU, we see that small amounts of rewinding require minimal additional computation, while more rewinding takes slightly longer due to instability slowing down convergence. For Lacuna-100, the cost of Algorithm 3 is small compared to the cost of learning and unlearning, due to a difference in batch size.

We note that the precomputation learning time of R2D is still vastly superior to that of the Hessian-Free method, the state-of-the-art second-order certified unlearning method. Moreover, by design, R2D (with checkpointing) will always cost less than retraining from scratch. This stands in contrast to CNS, which has a fixed computation cost that is not necessarily more efficient than retraining.

Table 8: Checkpoint reconstruction for eICU and Lacuna-100 datasets.

| eICU (136k parameters) | | Lacuna-100 (11 million parameters) | |
|---|---|---|---|
| Rewind Percent | Euclidean Distance | Rewind Percent | Euclidean Distance |
| 10.26% | 0.0650 | 7.41% | 1.174 |
| 23.08% | 0.2921 | 18.51% | 2.737 |
| 35.90% | 0.9977 | 25.92% | 3.187 |
| 48.71% | 2.924 | 44.44% | 4.464 |

Table 9: Learning times with checkpoint reconstruction (Algorithm 3) on the eICU dataset.

| Algorithm | Learning Time |
|---|---|
| R2D with checkpoint reconstruction, 23 % | 231.33 sec |
| R2D with checkpoint reconstruction, 49 % | 337.18 sec |

Table 10: Learning times with checkpoint reconstruction (Algorithm 3) on the Lacuna-100 dataset.

| Algorithm | Learning Time |
|---|---|
| R2D with checkpoint reconstruction, 19 % | 4.57 hours +123.62 sec |
| R2D with checkpoint reconstruction, 44 % | 4.57 hours +280 sec |

## B.4 LLM Experiments

We note that LLM unlearning is a rich subfield with its own specific experiment designs, evaluation metrics, and membership attacks. In this section, we provide a preliminary experiment to demonstrate the potential benefits of rewinding as opposed to "descending," i.e. continuing directly from the final model parameters, for LLMs. The code is available in the GitHub repository.

We finetune an open-source LLM (Mistral-7B-Instruct-v0.2 [2]) using LoRA [24] on the Alpaca dataset [43]. We treat the math-related tasks in Alpaca as the unlearned set $\mathcal{D}_{unlearn}$. We generate the following sets of model parameters:

1. Learned Model: We fine-tune the off-the-shelf model for three epochs on the whole Alpaca dataset.

2. "Retraining" Baseline: We fine-tune the off-the-shelf model for three epochs on only the non-math tasks. This simulates the "retrain from scratch" unlearning baseline.

3. "Finetuning" Baseline: We fine-tune for one more epoch starting from the Learned Model on the non-math tasks. This simulates the "finetuning" or D2D unlearning baseline. We note there is some overlap in terminology; in unlearning literature, "finetuning" refers to starting from the trained model and performing more steps on the loss function of the retained data.

4. R2D: We rewind to the first or second checkpoints from the Learned Model fine-tuning process, and proceed for two or one more epochs on the non-math tasks.

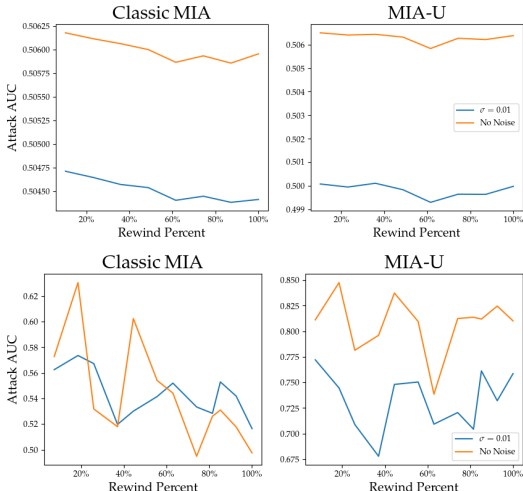

Figure 4: MIA scores for the eICU dataset (top) and the Lacuna-100 dataset (bottom).

In Table 11, we compare the performance (BLEU) of these models on $\mathcal{D}_{unlearn}$ and the computation cost of unlearning. We observe that both R2D methods outperform Finetuning, and in particular R2D 33% performs better while using the same amount of computation. This further supports our claim that rewinding is better than direct finetuning for unlearning on nonconvex functions.

| Algorithm | BLEU | Unlearning Epochs |
|---|---|---|
| Learned Model | 0.1576 | N/A |
| "Retraining" Baseline | 0.0935 | 3 |
| "Finetuning" Baseline | 0.1035 | 1 |
| R2D 33% | 0.1009 | 1 |
| R2D 67% | 0.1020 | 2 |

Table 11: Experiments with LoRA finetuning on an LLM.

## B.5    Additional Results

In this section, we provide additional experimental results, including full version of Tables 2 and 3 in Tables 12 and 14. For additional comparison purposes, we also provide metrics computed on the *noiseless* version of the certified unlearning methods in Tables 13 and 15. We include the standard deviation (1-sigma error) of the MIA scores over all $k$-fold cross-validation trials.

Figure 4 shows the MIA scores decreasing as the rewind percent increases, demonstrating that the attacks become less successful with more unlearning steps. This suggests that the rewinding step is important for erasing information about the data even without noise disguising the output of the unlearning algorithm. Nonetheless, Figure 4 also shows that a small perturbation consistently reduces the effectiveness of the membership attack. Finally, we observe that the MIA are generally more successful on the Lacuna-100 dataset than on the eICU dataset. This may be because the eICU dataset is noisier and the models are trained to a lower level of accuracy, leading to errors in classification that that are passed to the attack model.

Finally, we plot t-SNE feature representations of the model outputs in Figure 5 before and after unlearning, observing that the unlearned samples are indeed more dispersed after unlearning.

Table 12: Comparison of Unlearning Algorithms on the eICU Dataset

| Algorithm | Model AUC | | | | MIA AUC | |
|---|---|---|---|---|---|---|
| | $\mathcal{D}_{ood}$ | $\mathcal{D}_{retain}$ | $\mathcal{D}_{unlearn}$ | $\mathcal{D}_{test}$ | Classic MIA | MIA-U |
| **Original Models** | | | | | | |
| HF | 0.7365 | 0.7490 | 0.7614 | 0.7472 | | |
| CNS | 0.7519 | 0.7628 | 0.7860 | 0.7602 | | |
| R2D (with noise) | 0.7207 | 0.7329 | 0.7458 | 0.7316 | | |
| **Certified Methods** ($\sigma = 0.01$) | | | | | | |
| HF | 0.7361 | 0.7476 | 0.7587 | 0.7465 | $0.5078_{\pm 0.0260}$ | $0.5108_{\pm 0.0202}$ |
| CNS | 0.7517 | 0.7622 | 0.7848 | 0.7601 | $0.5145_{\pm 0.0283}$ | $0.5144_{\pm 0.0252}$ |
| R2D 10% | 0.7220 | 0.7335 | 0.7444 | 0.7327 | $0.5047_{\pm 0.0283}$ | $0.5001_{\pm 0.0255}$ |
| R2D 23% | 0.7220 | 0.7334 | 0.7442 | 0.7327 | $0.5046_{\pm 0.0283}$ | $0.5000_{\pm 0.0255}$ |
| R2D 36% | 0.7220 | 0.7334 | 0.7441 | 0.7327 | $0.5046_{\pm 0.0283}$ | $0.5001_{\pm 0.0255}$ |
| R2D 49% | 0.7219 | 0.7334 | 0.7441 | 0.7326 | $0.5045_{\pm 0.0283}$ | $0.4998_{\pm 0.0255}$ |
| R2D 74% | 0.7219 | 0.7333 | 0.7439 | 0.7326 | $0.5044_{\pm 0.0283}$ | $0.4997_{\pm 0.0255}$ |
| R2D 87% | 0.7218 | 0.7333 | 0.7438 | 0.7325 | $0.5044_{\pm 0.0283}$ | $0.4996_{\pm 0.0254}$ |
| R2D 100% | 0.7217 | 0.7331 | 0.7437 | 0.7323 | $0.5044_{\pm 0.0283}$ | $0.5000_{\pm 0.0253}$ |
| **Noiseless Retrain** (R2D 100%) | 0.7210 | 0.7335 | 0.7447 | 0.7321 | $0.5060_{\pm 0.0285}$ | $0.5064_{\pm 0.0241}$ |
| **Non-certified Methods** | | | | | | |
| Finetune | 0.7225 | 0.7352 | 0.7463 | 0.7337 | $0.5063_{\pm 0.0286}$ | $0.5066_{\pm 0.0248}$ |
| Fisher Forgetting | 0.7201 | 0.7310 | 0.7482 | 0.7302 | $0.5114_{\pm 0.0289}$ | $0.5102_{\pm 0.0221}$ |
| SCRUB | 0.7211 | 0.7336 | 0.7450 | 0.7322 | $0.5060_{\pm 0.0285}$ | $0.5056_{\pm 0.0247}$ |

Table 13: Certified Unlearning Algorithms without Noise ($\epsilon = \infty$) on the eICU Dataset

| Algorithm | Model AUC | | | | MIA AUC | |
|---|---|---|---|---|---|---|
| | $\mathcal{D}_{ood}$ | $\mathcal{D}_{retain}$ | $\mathcal{D}_{unlearn}$ | $\mathcal{D}_{test}$ | Classic MIA | MIA-U |
| **Original Models** | | | | | | |
| HF | 0.7365 | 0.7490 | 0.7614 | 0.7472 | | |
| CNS | 0.7519 | 0.7628 | 0.7860 | 0.7602 | | |
| R2D | 0.7211 | 0.7337 | 0.7451 | 0.7322 | | |
| **Certified Methods** ($\sigma = 0$) | | | | | | |
| HF | 0.7365 | 0.7490 | 0.7612 | 0.7472 | $0.5078_{\pm 0.0260}$ | $0.5135_{\pm 0.0195}$ |
| CNS | 0.7521 | 0.7630 | 0.7862 | 0.7604 | $0.5143_{\pm 0.0280}$ | $0.5138_{\pm 0.0254}$ |
| R2D 10% | 0.7214 | 0.7339 | 0.7452 | 0.7325 | $0.5062_{\pm 0.0286}$ | $0.5065_{\pm 0.0243}$ |
| R2D 23% | 0.7213 | 0.7338 | 0.7450 | 0.7324 | $0.5061_{\pm 0.0286}$ | $0.5064_{\pm 0.0244}$ |
| R2D 36% | 0.7213 | 0.7338 | 0.7450 | 0.7324 | $0.5061_{\pm 0.0286}$ | $0.5064_{\pm 0.0242}$ |
| R2D 49% | 0.7212 | 0.7338 | 0.7449 | 0.7324 | $0.5060_{\pm 0.0285}$ | $0.5063_{\pm 0.0244}$ |
| R2D 74% | 0.7212 | 0.7337 | 0.7449 | 0.7323 | $0.5059_{\pm 0.0285}$ | $0.5063_{\pm 0.0244}$ |
| R2D 87% | 0.7212 | 0.7337 | 0.7448 | 0.7323 | $0.5059_{\pm 0.0285}$ | $0.5062_{\pm 0.0244}$ |
| **Noiseless Retrain** (R2D 100%) | 0.7210 | 0.7335 | 0.7447 | 0.7321 | $0.5060_{\pm 0.0285}$ | $0.5064_{\pm 0.0241}$ |

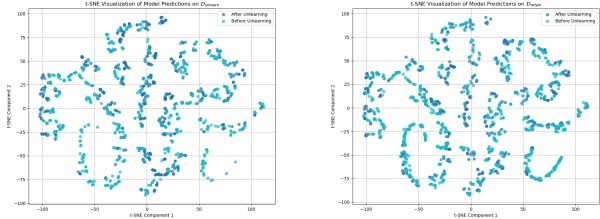

Figure 5: Left: t-SNE representations on $\mathcal{D}_{retain}$. Right: t-SNE representations on $\mathcal{D}_{unlearn}$.

Table 14: Comparison of Unlearning Algorithms on the Lacuna-100 Dataset

| Algorithm | Model AUC | | | | MIA AUC | |
|---|---|---|---|---|---|---|
| | $\mathcal{D}_{ood}$ | $\mathcal{D}_{retain}$ | $\mathcal{D}_{unlearn}$ | $\mathcal{D}_{test}$ | Classic MIA | MIA-U |
| **Original Models** | | | | | | |
| HF | 0.9392 | 0.9851 | 0.9746 | 0.9737 | | |
| CNS | 0.9497 | 0.9983 | 0.9956 | 0.9856 | | |
| R2D (with noise) | 0.9106 | 0.9719 | 0.9742 | 0.9591 | | |
| **Certified Methods ($\sigma = 0.01$)** | | | | | | |
| HF | 0.9327 | 0.9757 | 0.9555 | 0.9652 | $0.4949_{\pm 0.0381}$ | $0.8461_{\pm 0.0179}$ |
| CNS | 0.9475 | 0.9977 | 0.9949 | 0.9847 | $0.6023_{\pm 0.0346}$ | $0.8482_{\pm 0.0217}$ |
| R2D 7% | 0.9090 | 0.9606 | 0.9238 | 0.9478 | $0.5625_{\pm 0.0351}$ | $0.7721_{\pm 0.0211}$ |
| R2D 19% | 0.9120 | 0.9728 | 0.9407 | 0.9559 | $0.5735_{\pm 0.0347}$ | $0.7446_{\pm 0.0277}$ |
| R2D 26% | 0.8889 | 0.9313 | 0.8580 | 0.9266 | $0.5672_{\pm 0.0350}$ | $0.7089_{\pm 0.0278}$ |
| R2D 37% | 0.9055 | 0.9683 | 0.9269 | 0.9525 | $0.5197_{\pm 0.0354}$ | $0.6779_{\pm 0.0341}$ |
| R2D 44% | 0.9151 | 0.9752 | 0.9290 | 0.9621 | $0.5300_{\pm 0.0342}$ | $0.7481_{\pm 0.0245}$ |
| R2D 56% | 0.8897 | 0.9634 | 0.9215 | 0.9440 | $0.5414_{\pm 0.0342}$ | $0.7504_{\pm 0.0247}$ |
| R2D 63% | 0.9018 | 0.9594 | 0.8411 | 0.9426 | $0.5519_{\pm 0.0358}$ | $0.7092_{\pm 0.0290}$ |
| R2D 74% | 0.9082 | 0.9712 | 0.8998 | 0.9534 | $0.5333_{\pm 0.0342}$ | $0.7206_{\pm 0.0257}$ |
| R2D 81% | 0.8955 | 0.9648 | 0.9059 | 0.9463 | $0.5284_{\pm 0.0343}$ | $0.7043_{\pm 0.0213}$ |
| R2D 85% | 0.9010 | 0.9672 | 0.9072 | 0.9509 | $0.5529_{\pm 0.0336}$ | $0.7613_{\pm 0.0233}$ |
| R2D 93% | 0.9136 | 0.9701 | 0.8814 | 0.9528 | $0.5418_{\pm 0.0339}$ | $0.7322_{\pm 0.0216}$ |
| R2D 100% | 0.8932 | 0.9589 | 0.8868 | 0.9407 | 0.5164 | 0.7586 |
| **Noiseless Retrain** (R2D 100%) | 0.9406 | 1.0000 | 0.9460 | 0.9840 | $0.4974_{\pm 0.0343}$ | $0.8101_{\pm 0.0262}$ |
| **Non-certified Methods** | | | | | | |
| Finetune | 0.9403 | 0.9997 | 0.9975 | 0.9854 | $0.6013_{\pm 0.0335}$ | $0.8497_{\pm 0.0233}$ |
| Fisher Forgetting | 0.9398 | 0.9982 | 0.9986 | 0.9831 | $0.5814_{\pm 0.0342}$ | $0.8735_{\pm 0.0191}$ |
| SCRUB | 0.9391 | 0.9989 | 0.9999 | 0.9845 | $0.6179_{\pm 0.0331}$ | $0.8586_{\pm 0.0239}$ |

Table 15: Certified Unlearning Algorithms without Noise ($\epsilon = \infty$) on the Lacuna-100 Dataset

| Algorithm | Model AUC | | | | MIA AUC | |
|---|---|---|---|---|---|---|
| | $\mathcal{D}_{ood}$ | $\mathcal{D}_{retain}$ | $\mathcal{D}_{unlearn}$ | $\mathcal{D}_{test}$ | Classic MIA | MIA-U |
| **Original Models** | | | | | | |
| HF | 0.9392 | 0.9851 | 0.9746 | 0.9737 | | |
| CNS | 0.9497 | 0.9983 | 0.9956 | 0.9856 | | |
| R2D | 0.9391 | 0.9989 | 0.9993 | 0.9844 | | |
| **Certified Methods ($\sigma = 0$)** | | | | | | |
| HF | 0.9393 | 0.9852 | 0.9744 | 0.9738 | $0.4949_{\pm 0.0381}$ | $0.8360_{\pm 0.0185}$ |
| CNS | 0.9497 | 0.9984 | 0.9957 | 0.9857 | $0.5896_{\pm 0.0349}$ | $0.9674_{\pm 0.0084}$ |
| R2D 7% | 0.9307 | 0.9903 | 0.9677 | 0.9741 | $0.5727_{\pm 0.0333}$ | $0.8112_{\pm 0.0239}$ |
| R2D 19% | 0.9379 | 0.9992 | 0.9732 | 0.9828 | $0.6305_{\pm 0.0337}$ | $0.8475_{\pm 0.0229}$ |
| R2D 26% | 0.9179 | 0.9582 | 0.8833 | 0.9506 | $0.5318_{\pm 0.0398}$ | $0.7815_{\pm 0.0243}$ |
| R2D 37% | 0.9317 | 0.9959 | 0.9598 | 0.9777 | $0.5179_{\pm 0.0354}$ | $0.7961_{\pm 0.0265}$ |
| R2D 44% | 0.9368 | 0.9980 | 0.9518 | 0.9815 | $0.6024_{\pm 0.0335}$ | $0.8374_{\pm 0.0216}$ |
| R2D 56% | 0.9406 | 1.0000 | 0.9567 | 0.9860 | $0.5541_{\pm 0.0335}$ | $0.8096_{\pm 0.0245}$ |
| R2D 63% | 0.9260 | 0.9917 | 0.8870 | 0.9720 | $0.5442_{\pm 0.0336}$ | $0.7385_{\pm 0.0267}$ |
| R2D 74% | 0.9387 | 1.0000 | 0.9381 | 0.9856 | $0.4948_{\pm 0.0352}$ | $0.8125_{\pm 0.0262}$ |
| R2D 81% | 0.9391 | 0.9999 | 0.9486 | 0.9843 | $0.5261_{\pm 0.0348}$ | $0.8137_{\pm 0.0262}$ |
| R2D 85% | 0.9337 | 0.9987 | 0.9475 | 0.9827 | $0.5309_{\pm 0.0361}$ | $0.8119_{\pm 0.0253}$ |
| R2D 93% | 0.9389 | 0.9968 | 0.9385 | 0.9801 | $0.5177_{\pm 0.0352}$ | $0.8246_{\pm 0.0194}$ |
| **Noiseless Retrain** (R2D 100%) | 0.9406 | 1.0000 | 0.9460 | 0.9840 | $0.4974_{\pm 0.0343}$ | $0.8101_{\pm 0.0262}$ |

# C  Limitations

In this section, we discuss some limitations of our work. First, we observe that while theoretically we can achieve any level of noise and privacy by controlling $K$, the number of unlearning steps, the value of $\sigma$ decreases very slowly with more rewinding, as shown in Figure 1f. This stands in contrast to results in convex unlearning, where the noise may decay exponentially with $K$. See Appendix E for more discussion.

Second, since in practice full-batch gradient descent is too computationally demanding for all but the smallest datasets, we closely approximate this setting using mini-batch gradient descent with a very large batch size. This approximation and others (like estimation of $L$ and $G$) introduce some imprecision into the level of certification, although the general principle of achieving probabilistic indistinguishability via Gaussian perturbation still holds. In addition, the large batch size slows down training, as the optimizer takes a long time to escape saddle points and converge to well-generalizing minima. Moreover, because we require constant step size, we also lose the benefits of advanced optimization techniques such as Adam, momentum, or batch normalization, all of which improve convergence speed and model performance. Developing certified unlearning algorithms for stochastic gradient descent and adaptive learning rates is a future direction of our work.

Third, we note that in the experimental settings we consider, we only achieve reasonable model utility at very high levels of $\epsilon$, which may not be used in practice. This is true for all certified unlearning algorithms considered in this paper, as shown in Figure 1b and 1e. Therefore in this regime, $\epsilon$ just represents a relative calibration of privacy rather than a specific theoretical meaning.

Finally, we note that while certified unlearning methods do leverage Gaussian perturbation to defend against membership inference attacks successfully, this comes at a cost to model performance. The non-certified algorithms tend have higher MIA scores but also better model performance.

# D  Broader Impacts

This paper expands on existing work in machine unlearning, which has the potential to improve user privacy and remove the influence of corrupted, biased, or mislabeled data from machine learning models. However, additional investigation may be necessary to determine the effectiveness of these algorithms in the real world.

# E  Additional Discussion of Related Work

To review, our algorithm is a first-order, black-box algorithm that provides $(\epsilon, \delta)$ unlearning while also maintaining performance and computational efficiency. In this section, we provide an in-depth comparison our algorithmic differences with existing convex and nonconvex certified unlearning algorithms (Table 16). These algorithms all involve injecting some (Gaussian) noise to render the algorithm outputs probabilistically indistinguishable, whether it is added to the objective function, to the final model weights, or at each step of the training process.

We are interested in comparing the noise-unlearning tradeoff of each of these algorithms. The second column of Table 16 lists the standard deviation $\sigma$ in terms of $\epsilon$, $\delta$, and other problem parameters for the other certified unlearning algorithms. Our result is analogous to that of [34], due to algorithmic similarities such as weight perturbation at the end of training. However, the required noise in [34] decays exponentially with unlearning iterations, while our noise decreases more slowly with increasing $K$ for our algorithm. This difference is because [34] considers strongly convex functions, where trajectories are attracted to a global minimum.

The third column of Table 16 highlights an additional advantage our algorithm has over existing approaches. Our theoretical guarantees do not require a uniform bound on the model weights $||\theta|| \leq R$, nor does $\sigma$ depend on $R$, in contrast to [12] and [48]. Both [48, 12] require a bounded feasible parameter set, which they leverage to provide a loose bound on the distance between the retraining and unlearning outputs dependent on the parameter set radius $R$. However, this yields excessive noise requirements, with the standard deviation $\sigma$ scaling linearly or polynomially with $R$.

Our work shares similarities with [12], a first-order white-box algorithm that uses noisy projected gradient descent to achieve certified unlearning, leveraging the existence and uniqueness of the

| Algorithm | Standard Deviation of Injected Gaussian Noise | Bounded $\|\theta\| < R$? |
|---|---|---|
| Newton Step [23] | $\sigma = \frac{4LG^2\sqrt{2\log(1.5/\delta)}}{\lambda^2(n-1)\epsilon}$ | No |
| Descent-to-delete (D2D) [34] | $\sigma = \frac{4\sqrt{2}M(\frac{L-\lambda}{L+\lambda})^K}{\lambda n(1-(\frac{L-\lambda}{L+\lambda})^K)(\sqrt{\log(1/\delta)+\epsilon}-\sqrt{\log(1/\delta)})}$ | No |
| Langevin Unlearning [12] | No closed-form solution for $\sigma$ in terms of $\epsilon, \delta$ | Yes |
| Constrained Newton Step (CNS) [48] | $\sigma = \frac{\left(\frac{2R(PR+\lambda)}{\lambda+\lambda_{min}} + \frac{32\sqrt{\log(d/\rho)}}{\lambda+\lambda_{min}} + \frac{1}{8}\right)LR\sqrt{2\log(1.25/\delta)}}{\epsilon}$ | Yes |
| Hessian-Free unlearning (HF) [37] | $\sigma = 2\eta G\frac{\sqrt{2\log(1.25/\delta)}}{\epsilon}\zeta_T^{-U}$ | No |
| **Our Work (R2D)** | $\sigma = \frac{2mG \cdot h(K)\sqrt{2\log(1.25/\delta)}}{Ln\epsilon}$ | No |

Table 16: Comparison of certified unlearning algorithms and their noise guarantees. We denote $K$ as the number of unlearning iterates, $\lambda$ as the regularization constant or strongly convex parameter, $M$ as the Lipschitz continuity parameter, and $d$ as the model parameter dimension. In addition, $R$ is the parameter norm constraint. For [48], $P$ represents the Lipschitz constant of the Hessian, $\lambda_{min}$ represents the minimum eigenvalue of the Hessian, and $\rho$ represents a probability less than 1. For [37], $\zeta_T^{-U}$ is a constant dependent on the upper and lower bounds of the Hessian spectrum that grows with the number of learning iterates $T$ for nonconvex objective functions.

limiting distribution of the training process as well as the boundedness of the projection set. Their guarantee shows that the privacy loss $\epsilon$ decays exponentially with the number of unlearning iterates. However, $\sigma$, the noise added at each step, is defined implicitly with no closed-form solution. We can only assert that for a fixed number of iterations $K$, $\sigma$ must be at least $O(R)$ to obtain $\epsilon = O(1)$. Because $\sigma$ cannot be defined explicitly, it is difficult to implement this algorithm for a desired $\epsilon$. For example, when performing experiments for the strongly convex setting, which is a simpler mathematical expression, they require an additional subroutine to find the smallest $\sigma$ that satisfies the target $\epsilon$. As for the nonconvex setting, they state that "the non-convex unlearning bound... currently is not tight enough to be applied in practice due to its exponential dependence on various hyperparameters."

