# OpenReview forum: "Rewind-to-Delete: Certified Machine Unlearning for Nonconvex Functions"
_NeurIPS.cc/2025/Conference — NeurIPS 2025 poster_

### Official Review · Reviewer_xkYS · 2025-06-29

**Clarity:** 3
**Significance:** 3
**Originality:** 3
**Rating:** 4
**Confidence:** 2

**Summary:**

The paper proposes R2D (Rewind-to-Delete), a certified machine unlearning algorithm designed for the black-box setting, where model retraining is infeasible. The method operates by rewinding the model to an earlier checkpoint  $\theta _{T-K}$, re-optimizing only on the retained data $\mathcal{D}'$ , and injecting calibrated Gaussian noise to satisfy $\left( {\varepsilon ,\delta } \right)$-certified unlearning guarantees. When the checkpoint $\theta _{T-K}$ is unavailable, a proximal point method is used to reconstruct it from the final model state $\theta _{T-K}$.

**Questions:**

Please answer the questions in the Weaknesses. All questions need to be answered.

**Ethical Concerns:**

["NO or VERY MINOR ethics concerns only"]

**Final Justification:**

The rebuttal has address some of my main concerns. I have increased my rating, but without more visualization results, the effectiveness of the proposed method still may not be very clear. So I only increase the rating to 4.

**Limitations:**

yes

**Paper Formatting Concerns:**

Figure 1 is difficult to interpret. In particular, it is unclear why the classification performance on $\mathcal{D}_{unlearn}$ in Figure 1(g) increases as the rewind depth $K$ grows.

**Quality:**

2

**Strengths And Weaknesses:**

Strengths:
1. The method does not require access to internal training dynamics or full gradient history, making it deployable in practical, real-world scenarios.
2. The use of differential privacy allows the algorithm to offer $\left( {\varepsilon ,\delta } \right)$-certified unlearning guarantees, enhancing interpretability and accountability.
3. Algorithm 3 introduces a proximal fixed-point iteration to approximate historical checkpoints, enabling backward recovery without explicit training logs.

Weaknesses:
1. The paper should visualize feature representations (e.g., via t-SNE or PCA) to show how samples enjoy different levels of privacy and unlearning guarantees.
2. As Out-of-Distribution (OOD) samples are typically less sensitive to the presence or absence of specific training data, they may not be a reliable probe for evaluating unlearning success. The current experimental setup does not convincingly demonstrate the advantage of the proposed method.
3. The paper should include comparisons with more (e.g., DP-based) machine unlearning methods.
4. The proposed method finetunes from a checkpoint at step $T-K$, and empirical results suggest that performance improves when $K$ is large (i.e., closer to full retraining). However, the cost savings over retraining from scratch on $\mathcal{D}'$ are not clearly demonstrated. The paper should provide a concrete runtime and resource comparison in tasks where the model has converged, to justify practical efficiency.

---

> ### Author Rebuttal · Authors · 2025-07-29
>
> Thank you for your insightful review and considerate comments. In the following, we respond to your concerns.
>
> **The paper should visualize feature representations (e.g., via t-SNE or PCA) to show how samples enjoy different levels of privacy and unlearning guarantees.**
>
> We thank you for this suggestion. Due to NeurIPS rebuttal guidelines, we cannot provide any new plots. However, we have performed this visualization of the features, observing that the unlearned samples are indeed more dispersed after unlearning. We will include the results in our camera-ready version.
>
> **As Out-of-Distribution (OOD) samples are typically less sensitive to the presence or absence of specific training data, they may not be a reliable probe for evaluating unlearning success. The current experimental setup does not convincingly demonstrate the advantage of the proposed method.**
>
> We only use the OOD dataset for the membership inference attack (MIA), not for direct evaluation of model performance. We agree that unlearning should have little effect on the model performance on the OOD set, and this is supported by experimental data in Tables 8, 9, 10, and 11. In fact, **we leverage this low sensitivity to make our MIA more robust**. The MIA compares the output of the model on two datasets: the unlearned data, and "unseen" data not present in the training set. The attack is more successful if it determines that the first dataset is more sensitive to unlearning than the second. Typically, the standard practice is to use the test set as the second dataset. However, in our framework the test set may still contain data from *users* in the training data, and will still be affected by unlearning even if the specific test samples were not used during training. Therefore, we determined that the OOD dataset, constructed specifically from *users* not present in the training data, provides a more appropriate comparison. We note that while we refer to these samples as "OOD," they do not fall outside of the *classes* considered by the model and can still be reliably and accurately classified.
>
> Our other unlearning metric is performance on the unlearned dataset $\mathcal{D} _{unlearn}$ before and after unlearning. A decrease in performance  suggests the model is losing information about the data. For additional information about our unlearning metrics and membership attacks, see Section 4.1 and B.1.1.
>
> **The paper should include comparisons with more (e.g., DP-based) machine unlearning methods.**
>
> Our work already compares against all existing DP/certified methods for **nonconvex** unlearning. Theoretically, we compare against the Constrained Newton Step (Zhang et al., 2024), the Hessian-Free method (Qiao et al., 2025), and Langevin Unlearning (Chien et al., 2024), and we also review several convex methods, as highlighted in Table 1 and Appendix E. Experimentally, we compare against CNS (Zhang et al., 2024) and HF (Qiao et al., 2025). We do not compare against convex DP/certified unlearning methods because they are not designed for nonconvex functions like deep neural networks, and we do not compare against (Chien et al., 2024) because, as stated in their paper, their result is mostly of theoretical interest and is currently too loose to be implemented in practice.
>
> **The proposed method finetunes from a checkpoint at step $T-K$, and empirical results suggest that performance improves when $K$ is large (i.e., closer to full retraining). However, the cost savings over retraining from scratch on $\mathcal{D}'$ are not clearly demonstrated. The paper should provide a concrete runtime and resource comparison in tasks where the model has converged, to justify practical efficiency.**
>
> By construction, R2D with checkpointing will always be more efficient than retraining from scratch, because $K \leq T$. We provide a runtime comparison in Tables 6 and 7 in the Appendix. As expected, the runtime of R2D unlearning is directly proportional to the amount of rewinding steps, such that 20% rewinding takes 20% of the original training time to run, translating to 80% in computational savings. Our results in Table 2 suggest that even a small $K$ yields competitive performance.
>
> Tables 6 and 7 also show that R2D is more practical and efficient compared to HF and CNS. In particular, although HF unlearns quickly, it requires 22-88 times more compute time during learning, highlighting the benefits of black-box algorithms that do not require complicated data-saving procedures during training. As for memory requirements, our algorithm requires $O(d)$ storage, which beats the $O(d^2)$ and $O(nd)$ requirements of CNS and HF respectively (Table 1). We note that in practice, HF actually requires $O(ndT)$ storage during training to ultimately compute the unlearning matrix, where $T$ is the number of learning iterates. This is completely intractable for moderately sized neural networks and datasets, so instead we follow their lead and implemented a scaled-down version of their algorithm that only stores the vectors of the samples we are planning to unlearn, a small fraction of the whole dataset. Even with this concession, HF requires 1.5 GB of storage for the eICU dataset and 136GB for the Lacuna-100 dataset solely during training, whereas R2D only needs to store the model weights at a checkpoint and the final iterate ($2 \times 8$ MB for eICU and $2 \times 45$ MB for Lacuna-100). We will update our manuscript to include these concrete comparisons.
>
> **Figure 1 is difficult to interpret. In particular, it is unclear why the classification performance on $\mathcal{D} _{unlearn}$ in Figure 1(g) increases as the rewind depth $K$ grows.**
>
> Figure 1(a) and 1(g) show the utility tradeoff for various fixed $\epsilon$ and rewind depth $K$, based on equation (3). More rewinding $K$ means less noise is required to achieve the same unlearning guarantee $\epsilon$, improving the overall performance of the model, including the performance on $\mathcal{D} _{unlearn}$. This is why 1(a) and 1(g) are similar. Figure 1(g) specifically shows that for all these varying levels of utility, the performance on $\mathcal{D} _{unlearn}$ is lower after unlearning. We will improve the contrast on this figure to make the point clearer.
>
> In contrast, Figure 2 plots the performance for *constant* noise (Table 2), thereby isolating the impact of rewinding. We see that rewinding *decreases* the performance on $\mathcal{D} _{unlearn}$, while the performance on $\mathcal{D} _{train}$ and $\mathcal{D} _{test}$ remain relatively constant. This supports the argument that rewinding is a key component of our unlearning algorithm and erases information about $\mathcal{D} _{unlearn}$ in the model. We will improve the notation of Figure 1 and 2 to make this distinction clearer.
>
> Thank you again for your thoughtful review, and please let us know if you have any further concerns about the paper.

---

> > ### Author Response · Authors · 2025-08-05
> >
> > Dear Reviewer xkYS,
> >
> > As the author-reviewer discussion period is coming to a close, we wanted to check whether our responses have sufficiently addressed your concerns. If so, would you kindly consider updating your score? We are more than happy to address any further questions or concerns.
> >
> > Thank you again for your time and effort in reviewing our work!

---

> > ### Comment · Reviewer_xkYS · 2025-08-09
> > **Response to Authors**
> >
> > Thank you for your detailed response. The rebuttal has address some of my main concerns. I have increased my rating, but without more visualization results, the effectiveness of the proposed method still may not be very clear. So please include the results in the revised version.

---

### Official Review · Reviewer_7iuP · 2025-07-03

**Clarity:** 3
**Significance:** 1
**Originality:** 2
**Rating:** 4
**Confidence:** 3

**Summary:**

In this paper, the authors propose a first-order black-box unlearning algorithm. The key contributions of the paper are (i) a certified unlearning algorithm for non-convex loss functions. (ii) Theoretical unlearning guarantees; linear convergence and generalization guarantees for PL inequality satisfying loss functions. (iii) better performance than existing methods within a novel experimental framework.

**Questions:**

1. How does the accuracy of the proximal point method vary with the choice of K or η? Are there failure modes where it cannot approximate a valid earlier iterate?
2. For large neural networks, does the "offline" computation of recovering earlier checkpoints scale efficiently? Is there any empirical or theoretical justification for its tractability?
3. Since σ depends on L and G (which are estimated), how robust are the unlearning guarantees to inaccurate estimates? Is there a safe fallback?
4. Unlearning in Continual or Online Learning: Can R2D be applied to continually updated models, or does it assume static datasets during training and unlearning?

**Ethical Concerns:**

["NO or VERY MINOR ethics concerns only"]

**Final Justification:**

The authors provided sufficient justification in terms of results to help improve the paper and also address many concerns that were raised. They also addressed the concerns about novelty and after keen look into the analysis and the proofs, I am willing to increase my score.

**Limitations:**

Not many limitations. The only thing I notice is that there is no novelty in methods. The authors themselves acknowledge that they just combine well established ideas together in a different fashion to get R2D. That is the only limitation that I could see.

**Paper Formatting Concerns:**

few typos. but nothing major.

**Quality:**

2

**Strengths And Weaknesses:**

Strengths:
+ First certified unlearning algorithm for general nonconvex functions that is both first-order and black-box, addressing a key gap in the current literature.
+ The major strength of the paper comes from the analysis (specifically the unlearning guarantees, generalization guarantees) for nonconvex loss functions. This seems to be the major contribution.
+ The method performs better than other baseline methods chosen


Weaknesses:
- While the authors propose a proximal point method for black-box checkpoint reconstruction, the computational cost and convergence behavior of this approach are briefly addressed and may not be practical for large-scale deployments without further assumptions. A clearer breakdown of when this reconstruction becomes prohibitive would help support real-world adoption.
- The paper does an excellent job of analyzing trade-offs among ε, σ, and rewind steps K, but it doesn't offer practical guidance on selecting these parameters. This might hinder adoption in real-world pipelines that prioritize usability over theoretical bounds.
- The experiments are well-crafted, especially in the user-centric setting, but are focused on classification with binary labels. While not required, the paper would be stronger if it discussed generalizability of R2D beyond this setup, especially since the claim is a general first-order method for nonconvex functions.

---

> ### Author Rebuttal · Authors · 2025-07-27
>
> Thank you for your insightful review and considerate comments. In the following, we respond to your concerns.
>
> **While the authors propose a proximal point method for black-box checkpoint reconstruction, the computational cost and convergence behavior of this approach are briefly addressed and may not be practical for large-scale deployments without further assumptions. A clearer breakdown of when this reconstruction becomes prohibitive would help support real-world adoption.**
>
> Thank you for this suggestion. Based on your feedback, we implemented the proximal point method (Algorithm 3) and evaluated its ability to reconstruct prior checkpoints. We will include these numerical results and their plots in the camera-ready version of the paper.
>
> We measured the success by the Euclidean distance between the original checkpoint and the reconstructed checkpoint, observing that they are close for small $K$ but grow apart as we rewind past half the original training trajectory, due to compounding approximation errors that ultimately leads to instability in convergence. Taking into account the model dimension, our results show that we can quite faithfully reconstruct checkpoints for moderate $K$.
>
> | eICU (136k parameters)| | | Lacuna-100 (11 million parameters) | |
> |----------------|----------|---|----------------|----------|
> | Rewind Percent | Euclidean Distance |   | Rewind Percent | Euclidean Distance |
> | 10.26% | 0.0650 | | 7.41%| 1.174  |
> | 23.08%| 0.2921 | | 18.51%| 2.737 |
> | 35.90%| 0.9977 | | 25.92%| 3.187 |
> | 48.71%   | 2.924 | | 44.44%| 4.464 |
>
> **(Q1) How does the accuracy of the proximal point method vary with the choice of $K$ or $\eta$? Are there failure modes where it cannot approximate a valid earlier iterate?**
>
> In the above, we address how performance degrades as $K$ increases. The $\eta$ parameter is fixed, as it is the learning rate of the *training* algorithm (Algorithm 1). A small $\eta$ corresponds to more regularization, allowing the subproblem to remain computationally tractable.
>
> **(Q2) For large neural networks, does the "offline" computation of recovering earlier checkpoints scale efficiently? Is there any empirical or theoretical justification for its tractability?**
>
> We also evaluated the computational cost of Algorithm 3 and found that in most cases, reconstructing the checkpoint has a minor impact on the overall runtime. In the following, we reproduce Table 6 and 7 from the paper, now including the time it takes to reconstruct the given checkpoint into the learning time. For eICU, we see that small amounts of rewinding require minimal additional computation, while more rewinding takes slightly longer due to instability slowing down convergence. For Lacuna-100, the cost of Algorithm 3 is small compared to the cost of learning and unlearning, due to a difference in batch size.
>
> We note that the precomputation learning time of R2D is still vastly superior to that of the Hessian-Free method, the state-of-the-art second-order certified unlearning method. Moreover, by design, R2D (with checkpointing) will *always* cost less than retraining from scratch. This stands in contrast to CNS, which has a fixed computation cost that is not necessarily more efficient than retraining.
>
> Our results are also theoretically supported since $\eta \leq \frac{1}{L}$ implies that the subproblem becomes a tractable convex problem (Lemma A.1). Moreover, because we solve the subproblem with SGD, our algorithm is still a pure first-order method that beats second-order methods in memory storage.
>
> **Table 6: eICU dataset**
> | Algorithm  | Learning Time | Unlearning Time |
> | ----------------------------- | ------------- | --------------- |
> | R2D, 23 % | 230.76 sec | 58.48 sec |
> | R2D, 49 %  | 230.76 sec | 113.69 sec |
> | R2D with checkpoint reconstruction, 23 % |231.33 sec|  58.48 sec |
> |R2D with checkpoint reconstruction, 49 %|337.18 sec|113.69 sec |
> | Hessian-Free (HF) | 5.5 hours | 0.01 sec |
> | Constrained Newton Step (CNS) | 266.0 sec | 293.6 sec |
>
> **Table 7: Lacuna-100 dataset**
> | Algorithm | Learning Time | Unlearning Time|
> | ----------------------------- | ------------- | --------------- |
> | R2D, 19 %  | 4.57 hours | 0.91 hours |
> | R2D, 44 %  | 4.57 hours | 2.06 hours |
> |  R2D with checkpoint reconstruction, 19 % | 4.57 hours +123.62 sec |0.91 hours|
> | R2D with checkpoint reconstruction, 44 %  | 4.57 hours +280 sec |2.06 hours|
> | Hessian-Free (HF) | 4.23 days | 0.00 hours |
> | Constrained Newton Step (CNS) | 0.58 hours | 0.32 hours |
>
> **The paper does an excellent job of analyzing trade-offs among $\epsilon$, $\sigma$, and rewind steps $K$, but it doesn't offer practical guidance on selecting these parameters.**
>
> Following the guidelines in (Zhang et al., 2024), we suggest first choosing a $\sigma$ that preserves model utility (in our work, we use $\sigma = 0.01$), and a $K$ within the computational budget, such as $10$% of the training iterates. From there one can compute the level of privacy $\epsilon$ achieved. Our experiments for various $\sigma$, including $\sigma = 0$, (Table 9 and 11) suggest that a small amount of rewinding and/or noise performs well empirically. We will include this text in our paper.
>
> **The experiments are well-crafted, especially in the user-centric setting, but are focused on classification with binary labels. While not required, the paper would be stronger if it discussed generalizability of R2D beyond this setup, especially since the claim is a general first-order method for nonconvex functions.**
>
> R2D does generalize to any empirical risk minimization problem, and the number of classes does not meaningfully change the method or theoretical guarantees. Even though some other papers consider a multiclass setting, they uniformly sample i.i.d. data from the dataset to form the unlearning set, which is essentially equivalent to i.i.d. unlearning in a binary setting.
>
> We sought to develop a more realistic experiment framework for unlearning, with more complex dataset considerations. For example, we chose the Lacuna-100/VGG-Face2 dataset because it is a standard unlearning benchmark dataset consisting of individual "user data" that can be classified by global characteristics, using the MAAD-Face labels. We considered a multiclass problem (i.e. hair color or age) but found that the MAAD-Face labels were much less accurate and the classes were highly imbalanced. Designing a multiclass benchmark dataset that is compatible with our novel framework is a potential future direction of our research.
>
> **(Q3) Since $\sigma$ depends on $L$ and $G$ (which are estimated), how robust are the unlearning guarantees to inaccurate estimates? Is there a safe fallback?**
>
> Inaccurate estimates indeed introduce some imprecision in the theoretical guarantee. However, the benefit of certified unlearning is that the basic principle of ensuring differential privacy via Gaussian perturbation, which scales with the strength of the privacy guarantee $\epsilon$, still holds. Empirically, we observe that certified unlearning methods with even a small amount of Gaussian noise tend to outperform non-certified algorithms that do not use noise (Table 2).
>
> **(Q4) Unlearning in Continual or Online Learning: Can R2D be applied to continually updated models, or does it assume static datasets during training and unlearning?**
>
> R2D, and certified unlearning in general, applies to static datasets during training and unlearning. While we can perform sequential unlearning, and *adding* a fixed number of samples would yield a symmetric analysis, the existing theory is not equipped for a constant stream of new samples or any shifts in the underlying data distribution. Developing a certified unlearning framework for handling sequential tasks would be an interesting new direction of research.
>
> **Limitation: The only thing I notice is that there is no novelty in methods. The authors themselves acknowledge that they just combine well established ideas together in a different fashion to get R2D.**
>
> We feel that our work is sufficiently novel in both method and analysis, and that its strength lies in its simple and intuitive structure. Many unlearning algorithms utilize various forms of gradient descent and/or checkpointing, but developing the underlying theory to support them, for nonconvex functions, is highly nontrivial. For example, our work is inspired by Descent-to-Delete (D2D), a first-order certified unlearning algorithm for strongly convex functions. However, directly extending the D2D analysis to the nonconvex setting is *impossible*, since it relies on the existence of a unique global minimum, which (i) attracts training trajectories and (ii) remains in a small neighborhood when the underlying loss function is changed. Neither (i) nor (ii) hold for the general nonconvex setting, where we may only converge to a local minima or saddle point. Our novel insight is we leverage rewinding instead of "descending" to bring the unlearned model closer to the retrained model by reversing the divergence in training trajectories caused by the unlearning bias. This is a brand new algorithmic idea and an important contribution of our work.
>
> In our Prior Work section, we discuss existing algorithms that use some form of checkpointing and "rewinding" (Kurmanji et al., 2023), or gradient ascent (Jang et al., 2023). However, these works **do not have any theoretical guarantees** and rely solely on empirical unlearning metrics. For example, (Kurmanji et al., 2023) rewinds to a checkpoint of the unlearning process where the error on the forget set is "just right," so as to defend against MIAs, which is completely different from our usage and rationale. Moreover, although Algorithm 3 does involve gradient steps "up" the loss function, it is not gradient ascent but an implicit Euler step, which is theoretically supported (Section 2).
>
> Thank you again for your thoughtful review, and please let us know if you have any further concerns about the paper.

---

> > ### Comment · Reviewer_7iuP · 2025-08-05
> >
> > Thanks to the authors for patiently answering the questions raised. I have read the reviews by other reviewers and I have also read the rebuttal provided by the authors. Certainly, they do a good job at presenting facts. However, I still feel that the novelty aspect is not fully addressed well. From the response provided by the authors, it seems to me that the only novelty is building theoretical guarantee (which I understand is non-trivial). Given this, I retain my score.

---

> ### Author Response · Authors · 2025-08-05
>
> Dear Reviewer 7iuP,
>
> As the author-reviewer discussion period is coming to a close, we wanted to check whether our responses have sufficiently addressed your concerns. If so, would you kindly consider updating your score? We are more than happy to address any further questions or concerns.
>
> Thank you again for your time and effort in reviewing our work!

---

### Official Review · Reviewer_RBM5 · 2025-07-03

**Clarity:** 3
**Significance:** 2
**Originality:** 3
**Rating:** 4
**Confidence:** 3

**Summary:**

This paper introduces a novel certified unlearning framework that operates in a first-order, black-box manner. The authors provide a rigorous theoretical analysis of the trade-off between privacy, utility, and efficiency, and further explore the special case of Polyak–Łojasiewicz (PL) loss functions. Comprehensive empirical experiments demonstrate that the proposed algorithm outperforms existing methods in terms of convergence speed and unlearning guarantees.

**Questions:**

1. The authors focus on a first-order method, yet second-order approaches often offer faster convergence and stronger guarantees. Could you provide both theoretical discussion and empirical comparisons against a suitable second-order baseline to clarify the pros and cons?

2. How does the proposed algorithm scale to very large datasets or large language models? Including at least a preliminary experiment (e.g., fine-tuning a small Transformer) would help demonstrate practical feasibility.

3. Gaussian noise injection is central to your method. Please explain intuitively and formally why certified guarantees fail without this noise, and add ablation studies comparing performance with and without noise to highlight its impact on privacy and utility.

**Ethical Concerns:**

["NO or VERY MINOR ethics concerns only"]

**Final Justification:**

I would like to keep my score:

1. This paper needs more explanations for the noise added in the method, because it is the key contribution of the method

2. The experiments need to improve the presentation. And results in Table 2 are incremental.

3. This method is a novel unlearning method with guarantees based on first order. However, the novelty is not enough, it would be better to extend to second order version.

**Quality:**

3

**Strengths And Weaknesses:**

Strengths

1. The paper delivers solid proofs for the privacy-utility-efficiency trade-off, offering clear bounds and insights.
2. A wide range of benchmarks, datasets, and baselines are evaluated, showcasing the method’s superior performance.
3. The treatment of PL loss functions provides additional depth and highlights practical scenarios where the algorithm excels.

Weaknesses

1. The paper does not compare against second-order unlearning methods, leaving unclear whether the proposed first-order approach truly balances effectiveness and efficiency.

2. Scalability to very large datasets or modern large language models is not demonstrated.

3. The necessity and impact of Gaussian noise are stated but not empirically isolated.

---

> ### Author Rebuttal · Authors · 2025-07-29
>
> Thank you for your insightful review and considerate comments. In the following, we respond to your concerns.
>
> **The paper does not compare against second-order unlearning methods, leaving unclear whether the proposed first-order approach truly balances effectiveness and efficiency.**
>
> **(Q1) The authors focus on a first-order method, yet second-order approaches often offer faster convergence and stronger guarantees. Could you provide both theoretical discussion and empirical comparisons against a suitable second-order baseline to clarify the pros and cons?**
>
> We do compare against both existing second-order certified unlearning methods for nonconvex functions: Hessian-Free method (Qiao et al., 2025) and Constrained Newton Step (Zhang et al., 2024). Theoretically, we compare against both methods in Table 1, with additional discussion in Section 3 and Appendix E, including a more detailed review of the noise guarantees in Table 12. Beyond the main benefits of first-order algorithms, our guarantee has several theoretical advantages. We note that both second-order methods involve a single step and have *no complexity tradeoff*, requiring a fixed amount of computation that is not guaranteed to be more efficient than retraining and may not be converged to a stationary point or local minima. In contrast, both our unlearning and our utility guarantees can be improved by increasing the number of iterations. In addition, our noise bound scales inversely with the size of the dataset $n$, implying that unlearning a fixed number of samples for large datasets has less impact than for small datasets. In contrast, the noise in (Zhang et al., 2024) and (Qiao et al., 2025) is fixed for all dataset sizes.
>
> Empirically, we implement both methods as baselines in all experiments. We find that R2D outperforms both second-order methods in unlearning metrics (Table 2), is computationally efficient (Table 6 and 7), and exhibits a better privacy-utility tradeoff in practice (Figure 1). In particular, we found that the HF method requires expensive precomputation before unlearning, and the unlearning Newton step in (Zhang et al., 2024) may be *more* costly than retraining (Table 6).
>
> **Scalability to very large datasets or modern large language models is not demonstrated.**
>
> **(Q2) How does the proposed algorithm scale to very large datasets or large language models? Including at least a preliminary experiment (e.g., fine-tuning a small Transformer) would help demonstrate practical feasibility.**
>
> We note that LLM unlearning is a very rich subfield with its own specific experiment designs, evaluation metrics, and membership attacks, and as such extensive experiments may be outside the scope of our work, which focuses on theoretical unlearning guarantees. We now provide an experiment to demonstrate the potential benefits of rewinding as opposed to continuing directly from the current model parameters, for LLMs. The details and findings are as follows.
>
> We consider fine-tuning an open-source LLM (Mistral-7B-Instruct-v0.2) using LoRA on the Alpaca dataset. We treat the math tasks in Alpaca as the unlearned set $\mathcal{D} _{unlearn}$. We generate the following models:
>
> 1. Learned Model: We fine-tune the off-the-shelf model for three epochs on the whole Alpaca dataset.
> 2. "Retraining" Baseline: We fine-tune the off-the-shelf model for three epochs on only the non-math tasks. This simulates the "retrain from scratch" unlearning baseline.
> 3. "Finetuning" Baseline: We fine-tune for one more epoch starting from the Learned Model on the non-math tasks. This simulates the "finetuning" unlearning baseline. We note there is some overlap in terminology; in unlearning literature, "finetuning" refers to starting from the trained model and performing more steps on the loss function of the retained data.
> 4. R2D: We rewind to the first or second checkpoints from the Learned Model fine-tuning process, and proceed for two or one more epochs on the non-math tasks.
>
> In the following table, we compare the performance (BLEU) of these models on $\mathcal{D} _{unlearn}$ and the computation cost of unlearning. We observe that both R2D methods outperform Finetuning, and in particular R2D 33% performs better while using the same amount of computation. This further supports our claim that rewinding is better than direct finetuning for unlearning on nonconvex functions.
>
> | Algorithm             | BLEU       | Unlearning Epochs |
> |-----------------------|------------|-------------------|
> | Learned Model        | 0.1576     | N/A               |
> | "Retraining" Baseline | 0.0935     | 3                 |
> | "Finetuning" Baseline |0.1035 | 1                 |
> | R2D 33%               | 0.1009| 1                 |
> | R2D 67%               | 0.1020 | 2                 |
>
> **The necessity and impact of Gaussian noise are stated but not empirically isolated.**
>
> **(Q3) Gaussian noise injection is central to your method. Please explain intuitively and formally why certified guarantees fail without this noise, and add ablation studies comparing performance with and without noise to highlight its impact on privacy and utility.**
>
> We do perform ablation studies to empirically isolate the impact of Gaussian noise, but due to space constraints they are located in the Appendix. Tables 8 and 10 provide the full numerical results with noise, and Tables 9 and 11 provide the same results without noise. Figure 4 shows the MIA results for noiseless and noised R2D. As expected, the noiseless results exhibit higher model utility but the noised results exhibit better unlearning, indicated by lower MIA scores. At the same time, we note that for both versions, increased rewinding improves unlearning and reduces the MIA success, showing that this is a critical component of our approach.
>
> Unlearning may fail without noise because, as first addressed in (Guo et al., 2020), small differences in parameter space can leak information about data that was in the training set, even if the unlearned model parameters are approximately close to retraining. We require noise to achieve probabilistic indistinguishability and disguise the potential effects of the unlearned data, leveraging the properties of the Gaussian mechanism to quantify the privacy guarantee in terms of $\epsilon$ and $\delta$.
>
> Thank you again for your thoughtful review, and please let us know if you have any further concerns about the paper.

---

> > ### Author Response · Authors · 2025-08-05
> >
> > Dear Reviewer RBM5,
> >
> > As the author-reviewer discussion period is coming to a close, we wanted to check whether our responses have sufficiently addressed your concerns. If so, would you kindly consider updating your score? We are more than happy to address any further questions or concerns.
> >
> > Thank you again for your time and effort in reviewing our work!

---

> > ### Comment · Reviewer_RBM5 · 2025-08-06
> >
> > Thank you for your response. I have seen results in Table 2, but it seems difficult to understand:
> >
> > 1. What’s the model performance (AUC)?
> > 2. Under “original models”, why there are several unlearning methods?
> > 3. How to distinguish the results of before unlearning and after unlearning? I think maybe “original models” part is before unlearning. If so, the table needs to adapt to show “unlearned models”.
> >
> > I think it would be better to improve this important table for readers.

---

> ### Author Response · Authors · 2025-08-07
>
> 1. Model performance (AUC) is the success of the model at performing the underlying classification task (detailed in "Datasets and Models" in Section 4.1), where AUC refers to the Area Under the Receiver Operating Characteristic Curve. One of our unlearning metrics is the decrease in AUC on $\mathcal{D} _{unlearn}$ before and after unlearning (Table 2). The other unlearning metric is MIA score (Table 3).
> 2. The "original models" is indeed the performance of the trained models before unlearning. There are different methods because the certified baseline methods (HF, CNS) require different training procedures. The non-certified baseline methods do not need special training procedures and are implemented on the R2D trained "original model."
> 3. In Table 2, "Original Models" refers to the models before unlearning, and all other rows ("Certified Methods," "Noiseless Retrain," and "Non-certified Methods") are all unlearned models. We will add an additional label of "Unlearned Models" to make this distinction clearer.
>
> Thank you for your feedback. The above information is included in Section 4, with some additional details in the Appendix. We will emphasize this information in the caption of Table 2 so the setup is especially clear.

---

> > ### Comment · Reviewer_RBM5 · 2025-08-07
> >
> > Thank you for the clarifications about the table 2. I will keep my score.

---

### Official Review · Reviewer_PXQd · 2025-07-04

**Clarity:** 2
**Significance:** 2
**Originality:** 2
**Rating:** 3
**Confidence:** 2

**Summary:**

This paper proposes a black-box, first-order machine unlearning algorithm for non-convex, Lipschitz, and smooth loss functions. The (randomized) unlearning algorithm takes as input the original model parameters, the full training dataset, and a subset of the training data to be unlearned. The goal is to return model parameters that are indistinguishable from those that would have been obtained had the subset never been part of the training set.
Effectively, this means the influence of the removed data on the model is significantly reduced.

The proposed algorithm uses only first-order information and analyzes the privacy-utility trade-off. Furthermore, when the non-convex loss satisfies the Polyak-Łojasiewicz (PL) inequality, a well-known condition in nonconvex optimization, a generalization guarantee is provided.

**Questions:**

See the above.

**Ethical Concerns:**

["NO or VERY MINOR ethics concerns only"]

**Limitations:**

Limitations are addressed.

**Quality:**

2

**Strengths And Weaknesses:**

Strengths:

The problem of certified machine unlearning for non-convex loss functions is important and underexplored.
This paper appears to make meaningful progress in this direction.

Weaknesses and Questions:

As someone familiar with the general concept of machine unlearning but not deeply familiar with previous algorithms:
How do the proposed privacy-utility and generalization guarantees compare to prior work?
This comparison is unclear from the current presentation.

The algorithm appears relatively simple. What are the novel contributions or insights compared to existing methods?

Why is the method described as a "black-box" algorithm?
Specifically: What is the initial point $\theta_0$? Is it the model trained on the full dataset, or some arbitrary initialization? If  $\theta_0$ is just an arbitrary initialization, in what sense is the method "black-box"?

What is the role of the parameter $K$ in the algorithm?

I think some clarifications are missing in the paper, which are necessary to assess the significance of the results.

---

> ### Author Rebuttal · Authors · 2025-07-29
>
> Thank you for your review and considerate comments. In the following, we respond to your concerns.
>
> **As someone familiar with the general concept of machine unlearning but not deeply familiar with previous algorithms: How do the proposed privacy-utility and generalization guarantees compare to prior work? This comparison is unclear from the current presentation.**
>
> We provide a list and detailed comparison of the theoretical guarantees of prior work in Table 12 in Appendix E. Our algorithm inherits the same basic $\sigma = O(\frac{1}{\epsilon})$ relationship as other differentially private (and certified unlearning) algorithms, since it follows directly from the Gaussian mechanism. However, computational efficiency is also a central concern for unlearning, which is why we also have a tradeoff with complexity, represented by the number of unlearning iterates $K$. Our algorithm and theoretical analysis allows flexible selection between the variables $\epsilon, \sigma$, and $K$, such that one can be chosen to be arbitrarily small compared to the other two. In particular, when we rewind to the initial iterate ($K = T$), we recover *perfect* privacy-utility, corresponding to full retraining and zero noise, at the expense of computation.
>
> Our privacy-utility-complexity guarantee (3) has several advantages over prior work in nonconvex certified unlearning, which we highlight in Section 3 and discuss further in Appendix E. First, (Zhang et al., 2024) and (Qiao et al., 2025) have *no complexity tradeoff*, both require a fixed amount of computation that is not guaranteed to be more efficient than retraining. In fact, we found in our experiments (Table 6) that the Newton step in (Zhang et al., 2024) may be *more* costly than retraining. Second, our noise bound scales inversely with the size of the dataset $n$, implying that unlearning a fixed number of samples for large datasets has less impact than for small datasets. In contrast, the noise in (Zhang et al., 2024) and (Qiao et al., 2025) is fixed for all dataset sizes. Finally, (Zhang et al., 2024) and (Chien et al., 2024) require projection onto a bounded parameter set, with fixed radius $R$, in order to control the training trajectory and thereby bound the distance between unlearning and retraining. As a result, their noise guarantees scale as $O(R)$, which is impractical and not necessarily better than randomly sampling weights from the feasible set.
>
> As for the generalization guarantee, we note that we can only obtain such guarantees for algorithms that converge to global minima. By considering PL functions, we are, to the best of our knowledge, the first to consider unlearning generalization in a nonconvex setting. However, we can compare our result to existing results for strongly convex functions, which are a subset of PL functions (Qiao et al., 2025; Liu et al., 2023; Suriyakumar et al., 2022). We observe that the population risk can be decomposed to three terms: an $O(\frac{1}{n})$ term representing the difference with empirical risk, an $O(\sqrt{d} \sigma)$ term representing the Gaussian noise, and a term representing the convergence of the underlying optimization framework. The first two terms are roughly equivalent across most works because they follow from fundamental principles. As for the last term, in our work it decays exponentially with increasing iterations, whereas (Liu et al., 2023) and (Suriyakumar et al., 2022) feature fixed bounds that cannot be reduced. Our result is roughly equivalent to Equation (14) of (Qiao et al., 2025) (since $T =  \frac{n E}{|B|}$). However, PL is a significantly weaker condition than strong convexity that cannot take advantage of a *unique* global minimum; instead, we show convergence by noting that gradient descent steps on the original loss function $f_{\mathcal{D}}$ represent biased gradient descent steps on the updated loss function $f_{\mathcal{D}'}$ (Corollary A.5). We will expand our discussion of the generalization guarantee in our camera-ready version.
>
> **The algorithm appears relatively simple. What are the novel contributions or insights compared to existing methods?**
>
> Beyond the main contributions of our work (first-order, black-box, nonconvex), **our algorithm addresses an important gap in machine unlearning.** There are two well-known algorithms in machine unlearning: Descent-to-Delete (D2D), a certified unlearning algorithm for strongly convex functions, and Finetuning, the non-certified analog to D2D. In particular, Finetuning is an extremely common baseline method implemented in almost every machine unlearning paper, including both theory and application papers. These algorithms are similar-- for both, the model is trained with (stochastic) gradient descent steps during the learning stage, and for unlearning, additional steps are performed starting from the trained model, on the retained dataset. However, neither have theoretical guarantees for nonconvex functions. In fact, it is likely *impossible* to extend the D2D analysis to the nonconvex setting, since it relies on the existence of a unique global minimum, which (i) attracts training trajectories and (ii) remains in a small neighborhood when the underlying loss function is changed. Neither (i) nor (ii) hold for the general nonconvex setting, where we may only converge to a local minima or saddle point.
>
> In our work, we argue that instead of performing steps starting from the trained model, one should "rewind" to an earlier checkpoint and perform steps starting from there. Our novel insight is we leverage rewinding instead of "descending" to bring the unlearned model closer to the retrained model by reversing the divergence in training trajectories caused by the unlearning bias. This is a brand new algorithmic idea and an important contribution of our work. We provide a clean theoretical analysis analyzing the privacy, utility, and complexity of this approach, showing that it is provably more efficient than retraining. Our work provides a theoretical basis for first-order methods in unlearning, but suggests that rewinding instead of fine-tuning is a better choice on nonconvex functions. We feel that the simplicity is not a flaw of our algorithm, rather it is part of its appeal, making it easier to understand and implement.
>
> **Why is the method described as a "black-box" algorithm? Specifically: What is the initial point $\theta_0$?**
>
> Our *unlearning* algorithm $U$ is black-box, which we define to be an unlearning algorithm that can be applied to a model pretrained with gradient descent, without any special procedures during learning. This stands in contrast to white-box algorithms, such as (Qiao et al., 2025), which requires expensive storage of information during training, and (Chien et al., 2024), which requires noise added at every training iteration. $\theta_0$ is the model initialization for the *learning* algorithm. We show that our *unlearning* algorithm is $\epsilon, \delta$-indistinguishable to the output of the learning algorithm $A$, if $A$ initialized at the same point $\theta_0$ and performed on $\mathcal{D}'$.
>
> **What is the role of the parameter $K$ in the algorithm?**
>
> $K$ is the number of unlearning iterates, where $T$ is the number of training iterates and we start the unlearning process from the $T - K$th training checkpoint. Informally, it is "the number of steps to go back," starting from the trained model $\theta_T$. We can "go back" or rewind either by directly loading an earlier checkpoint or reconstructing one via Algorithm 3.
>
> We will improve the manuscript by making the above points clearer in the camera-ready version. Thank you again for your thoughtful review, and please let us know if you have any further concerns about the paper.

---

> > ### Comment · Reviewer_PXQd · 2025-08-05
> >
> > Thanks for your detailed response.
> > From the perspective of a non-expert in the field, it's quite difficult to understand the key aspects of the proposed method from the main body of the paper, which also makes it hard to assess the novelty. At least in terms of presentation, I don't think the paper is ready for publication.
> > Since I'm not deeply familiar with other methods in the literature, I will defer to the other reviewers. If they support acceptance, I won't object.
> > For now, I will keep my score as is.

---

> > > ### Author Response · Authors · 2025-08-08
> > >
> > > **At least in terms of presentation, I don't think the paper is ready for publication.**
> > >
> > > We respectfully disagree. We invite the Area Chair to personally evaluate the clarity of our presentation and proposed methods (in brief, the abstract, Table 1, and Algorithm 1, 2, and 3).

---

> ### Author Response · Authors · 2025-08-05
>
> Dear Reviewer PXQd,
>
> As the author-reviewer discussion period is coming to a close, we wanted to check whether our responses have sufficiently addressed your concerns. If so, would you kindly consider updating your score? We are more than happy to address any further questions or concerns.
>
> Thank you again for your time and effort in reviewing our work!

---

### Note · Authors · 2025-08-11

Thank you to all reviewers and the Area Chair for their time and effort in reviewing our work. Rather than directly repeat what is stated in the paper and rebuttals, we want to address two key points about the *broader context* of our work, that perhaps were not immediately obvious from our paper.

First, although we compare against other certified unlearning/DP algorithms in our paper, what may be less obvious is that almost all these methods boil down to combinations of empirical risk minimization and Gaussian perturbation. This a well-established theme of both early DP works and more recent certified unlearning works (see response to Reviewer 7iuP for some examples). While these algorithms may seem similar, small differences lead to completely different analyses and theoretical guarantees. In this context, our work matches (and in some cases, exceeds) the **novelty** of existing work in algorithm, analysis, and experimental design.

Second, while we do compare our work to "Descent-to-Delete," what may be less obvious is that D2D is an important algorithm because it is the basis for the standard "Finetuning" baseline method which is implemented in virtually all unlearning papers, including theory and application works. This is usually because finetuning is first-order and easy to implement, even though D2D only has theoretical guarantees on strongly convex functions (which are impossible to extend to nonconvex settings) and finetuning does not perform well empirically on deep neural networks. In contrast, R2D is *equally* easy to implement, has theoretical guarantees on nonconvex functions, and empirically outperforms certified and non-certified methods. Therefore, we don't just propose a highly performant unlearning method, but we also provide strong support that rewinding instead of "descending" is a more appropriate baseline method for comparison.

In our work, we provide a robust review of our theoretical and technical contributions compared to existing certified unlearning works, but perhaps without sufficiently emphasizing the above two points regarding **novelty** and **impact**. We will revise the manuscript to include the contribution of our work in the context of machine unlearning and DP as a whole.

---

### Decision · Program_Chairs · 2025-09-17

**Decision:**

Accept (poster)

**Comment:**

This paper proposes a first-order, black-box unlearning algorithm for non-convex loss problems. The proposed algorithm works by rewinding to an earlier model checkpoint during learning and then performing gradient descent on the retain dataset. They prove the certified unlearning guarantee of the proposed algorithm and provide the privacy-utility-complexity trade-off. They also provide a generalization bound for PL problems. Finally, they test their proposed algorithm in an empirical evaluation, showing its effectiveness.

**Strengths**
- The problem of certified machine unlearning for non-convex loss functions is important and underexplored
- The idea of rewinding for certified unlearning, while embarrassingly simple, is interesting and practically relevant
- The proposed algorithm has theoretical guarantees on nonconvex functions, and empirically outperforms certified and non-certified methods in a wide range of benchmarks, datasets, and baselines.
- The method does not require access to internal training dynamics or full gradient history, making it deployable in practical, real-world scenarios. Especially, Algorithm 3 introduces a proximal fixed-point iteration to approximate historical checkpoints, enabling backward recovery without explicit training logs.


**Weaknesses**
- The writing somewhat makes it not straightforward to understand, as pointed out by Reviewer PXQd. For example, the role of $K$ and the notion of "back-box" have not been elaborated well enough in the writing.
- Some concern about the technical novelty of the paper given the algorithm is simple, but I think the authors have done a fairly good job of positioning their contribution in a broader context (see the author response in comparison with the D2D algorithm, when responding to Reviewer PXQd)

There were mixed feedbacks initially among the reviewers. After the rebuttal, the authors have successfully convinced two reviewers to turn into the acceptance front.